# Methylation reprogramming associated with aggressive prostate cancer and ancestral disparities

Jenna Craddock [1,2], Pavlo Lutsik [3,4], Pamela X Y Soh [2], Melanie Louw [5,6], Md Mehedi Hasan [2], Sean M Patrick [1], Shingai B A Mutambirwa [7], Phillip D Stricker [8], Hagen E A Förtsch[9] & , HEROIC PCaPH Africa1K*, M S Riana Bornman [1], Clarissa Gerhäuser [3] & Vanessa M Hayes [1,2,10,11] ✉

## Abstract

**African men are disproportionately impacted by aggressive prostate cancer (PCa). The key to this disparity is both genetic and environmental factors, alluding to epigenetic modifications. However, African-inclusive prostate tumour DNA methylation studies are lacking. Assembling a multi-geo-ancestral prostate tissue cohort, including men with (57 African, 48 European, 23 Asian) or without (65 African) PCa, we interrogate for genome-wide differential methylation. Overall, methylation appears to be driven by ancestry over geography (152 southern Africa, 41 Australia). African tumours show substantial heterogeneity, with universal hypermethylation indicating more pervasive epigenetic silencing, encompassing PCa suppressor genes and enhancer-targeted binding motifs. Conversely, African tumour-associated heterochromatic hypomethylation suggests chromatin relaxation and developmental pathway activation via enhancer targets. Notably, non-prostate lineage elements appeared preferentially exploited in African tumorigenesis, with ancestry potentially influencing the extent of lineage-inappropriate activation, and tumour progression marked by repression of developmental regulators. Together, these findings point to extensive epigenetic plasticity in African tumours, with intergenic regulatory remodelling promoting genomic instability, metastatic potential and aggressive disease phenotypes.**

**Keywords** Prostate Tumours; Differential Methylation; African Ancestry; Health Disparity
**Subject Categories** Cancer; Chromatin, Transcription & Genomics

## Introduction

Prostate cancer (PCa) is a significant global health concern. The second most frequently diagnosed cancer among men worldwide, PCa was responsible for ~1.47 million new cases and 397,000 deaths in 2022 (Bray et al, 2024). However, PCa disproportionately burdens men of African ancestry and/or from Africa. A focus on the United States has revealed that African American men display a 2- and 4.3-fold increased risk for PCa mortality compared to their European and Asian counterparts, respectively (Surveillance Research Program National Cancer Institute, 2024). Conversely, Sub-Saharan Africa has received the top four of five ranks for global PCa mortality rates, with southern Africa leading the charge at a 4-fold increase (Bray et al, 2024). This disparity underscores the urgent need for research efforts tailored to the African context. Yet, Sub-Saharan African populations remain underrepresented in PCa research (Samtal et al, 2022), with limited focus on exomic (White et al, 2022) and whole genome interrogation (Jaratlerdsiri et al, 2022, 2018). The role of epigenetics remains unexplored.

DNA methylation aberrations are some of the earliest, most stable and frequent molecular changes to occur in PCa (Lam et al, 2020). More frequent than genetic alterations, they are preserved throughout progression and metastatic development. With no known significant PCa modifiable risk factors (Kensler and Rebbeck, 2020), somatic DNA methylation profiling holds potential to identify contributing environmental carcinogens (Mikeska and Craig, 2014). Notable DNA methylation differences in prostate tumours between ancestries include *TIMP3* hypermethylation in Black over White Americans (Rubicz et al, 2019), while glaring inconsistencies include prostate tumour *CD44* hypermethylation (Woodson et al, 2003) versus no differential methylation (Das et al, 2006), and *PMEPA1* hypermethylated (Rubicz et al, 2019) versus hypomethylated (Sharad et al, 2014) for Black Americans. Well-established for being epigenetically silenced in PCa, *GSTP1* appears to be universally hypermethylated across prostate tumours regardless of patient ancestry (Enokida et al, 2005). While methylation profiles distinguishing prostate tumour from benign tissue appear

[1]School of Health Systems and Public Health, Faculty of Health Sciences, University of Pretoria, Pretoria, South Africa. [2]Ancestry and Health Genomics Laboratory, Charles Perkins Centre, School of Medical Sciences, Faculty of Medicine and Health, University of Sydney, Camperdown, NSW, Australia. [3]Division of Cancer Epigenomics, German Cancer Research Center (DKFZ), Heidelberg, Germany. [4]Department of Oncology, KU Leuven, Leuven, Belgium. [5]National Health Laboratory Services, Johannesburg, South Africa. [6]Department of Anatomical Pathology, School of Pathology, University of the Witwatersrand, Johannesburg, South Africa. [7]Department of Urology, Sefako Makgatho Health Science University, Dr George Mukhari Academic Hospital, Medunsa, Ga-Rankuwa, South Africa. [8]Department of Urology, St Vincent's Hospital and Private Clinic, Darlinghurst, NSW, Australia. [9]Windhoek Central Hospital, University of Namibia, Hage Geingob Campus, Windhoek Khomas, Namibia. [10]Manchester Cancer Research Centre, University of Manchester, Manchester, UK. [11]Norwich Medical School, University of East Anglia, Norwich, UK. *Lists of authors and their affiliations appear at the end of the paper. ✉E-mail: vanessa.hayes@sydney.edu.au

to be similar across American ancestries (Chernoff et al, 2022), dysregulation of the androgen receptor signalling pathway coupled with decreased expression of immune-related genes has been reported to be African-derived tumour specific (Ramakrishnan et al, 2024). Inconsistency between studies likely arises from small sample sizes, varying methylation profiling methods, non-biological factors or the heterogenous nature of populations investigated (Stevens et al, 2023), including historical genetic admixture (Zakharia et al, 2009). Data for Sub-Saharan Africa appears to be lacking.

In addition, European bias associated with DNA methylation profiling array design, including the Illumina MethylationEPIC BeadChips, raises concerns with regard to poor CpG probe hybridisation within genetically highly diverse African populations (Zhou et al, 2017). Recognising this caveat, Zhang et al (Zhang et al, 2022) recently characterised Illumina probes in global populations using the 1000 Genomes Project data. Including East and West African, and predominantly West African ancestral Black American and African Caribbean populations (Zakharia et al, 2009), southern Africans at greatest risk for PCa mortality (Bray et al, 2024) and representing the greatest regional genetic diversity (Choudhury et al, 2017), are yet to be represented.

In this study of prostate tissue derived from a unique cohort of 193 individuals biased towards southern Africans (122 African, 30 European), with further comparison to Australians (18 European, 23 Asian), we generate not only a southern African-relevant methylome-wide EPIC array filtering resource, but also interrogate for both geo-ancestral and tissue-specific differential DNA methylation. We find prostate tumour methylation to be driven by patient ancestry over geography, while African tumours show significant hypermethylation, heterogeneity and elevated gene silencing. The identification of African-specific PCa gene targets, as well as greater African-derived tumour immune function impairment, provides further rationale for the importance of African inclusion in epigenetic studies focused on reducing the global PCa burden.

# Results

## African-inclusive DNA methylation array limitations, confounders and cohort characterisation

Avoiding unreliable hybridisation with consequential DNA methylation mismeasurement (Zhou et al, 2017), single-nucleotide polymorphism (SNP)-overlapping probe filtering is a common data processing feature in current workflows. Established using largely European-derived resources, although a more recent African-inclusive strategy has been proposed (Zhang et al, 2022), our focus on genetically diverse southern Africans raises concerns. Using published whole genome sequenced blood-derived data (average coverage 43×; range 32–69×) from 99 genetically confirmed southern Africans, with germline single-nucleotide variants (SNVs) and insertions/deletions <50 bases (indels) called referencing GRCh38 (Jaratlerdsiri et al, 2022), we generate a southern African-relevant variant filtering resource (minor allele frequency (MAF) > 0.01), comprising 56,280 SNVs and 3623 indels (EPICv1, Datasets EV1–EV3), and 50,786 SNVs and 3087 indels (EPICv2, Datasets EV4–EV6). Our workflow is outlined in Fig. EV1.

Using our SNP-filtering resource, we further evaluate for African-specific discrepancies in probe content between EPIC array versions. Observing a greater proportional loss for EPICv1 (6.91% or 59,903) over EPICv2 sites (5.75% or 53,873), SNP overlap was identified in three categories (MAF > 0.01): (i) target CpG sites (45,669 EPICv1; 33,772 EPICv2), (ii) single base extension (SBE) sites of Type I probes (2359; 746), and (iii) overlapping the probe body within 5 bp of the target CpG (22,103; 25,029). Our African EPICv1 assessment reveals a 1.4-fold overlap with target CpG sites, and 1.6-fold with SBE sites when compared with equivalent European-relevant confounding (Pidsley et al, 2016). Due to African variant confounding and compared with EPICv1, EPICv2 displays greater proportional loss within gene bodies (49.50% EPICv1; 59.14% EPICv2) (Fig. 1A), non-CpG island (CGI) regions (72.97%; 79.08%) (Fig. 1B) and across Type II probes (84.23%; 89.49%) (Fig. 1C) and FANTOM5 enhancers (3.80%; 3.98%) (Fig. 1D). This observation is likely explained by elevated coverage over said regions and greater Type II probe content (abundant over Type I across platforms) on EPICv2 over EPICv1 (Peters et al, 2024). Within gene regions, although probe distribution across platforms is dominant in transcription start sites (TSSs) (Peters et al, 2024), probe loss does not reflect this distribution—a positive finding.

To evaluate cross-platform reproducibility, we profiled a subset of matched African sample pairs on both the EPICv1 and EPICv2 array ($n = 7$), observing a high β-value correlation between the EPIC arrays (all $\rho > 0.9838$) (Fig. 1E; Appendix Fig. S1). Suggesting African-specific reproducibility and suitability for cross-platform integration, it was surprising that our 193-sample merged dataset (78 EPICv1; 115 EPICv2) (Table 1) showed, after normalisation and background subtraction, significant array version confounding with principal component analysis (PCA) (Fig. 1F; Dataset EV7). We acknowledge that minor variation may arise from lot-to-lot differences in array manufacture. However, we note that this array-driven separation was captured by PC2, which explained only 5.03% of total variance, whereas PC1, accounting for 59.18%, was strongly associated with ancestry, suggesting that biological rather than technical factors dominate overall variability. Notably, this array-driven separation cannot adequately be explained by possible variation across different sample collection sites. Using our genetically confirmed through ancestral fraction analyses (122 African, 48 European, 23 Asian; Fig. 1G) and geographically (148 South Africa, 4 Namibia, 41 Australia) diverse cohort to eliminate this confounding contribution, we approached our analyses using a larger discovery versus smaller validation cohort derived from alternate array types (Table 1). As such, our "ancestry-associated" (African versus non-African) analysis included an EPICv1 discovery ($n = 70$) and EPICv2 validation cohort ($n = 57$), while our "tumour-associated" (tumour versus normal/benign tissue) analysis included an EPICv2 discovery ($n = 93$) and EPICv1 validation cohort ($n = 29$). We present only discovery cohort observations that were consistent with the validation cohort.

Overall, we observed vast DNA methylation heterogeneity amongst our African samples. Notably, seven African samples histologically classified as "non-tumour" revealed DNA methylation profiles resembling tumour samples (Appendix Fig. S2). Speculating that cancerous tissue may have been missed during routine biopsy, these samples were recharacterized as "presumably PCa" in downstream analyses. In addition, as we observed β-value

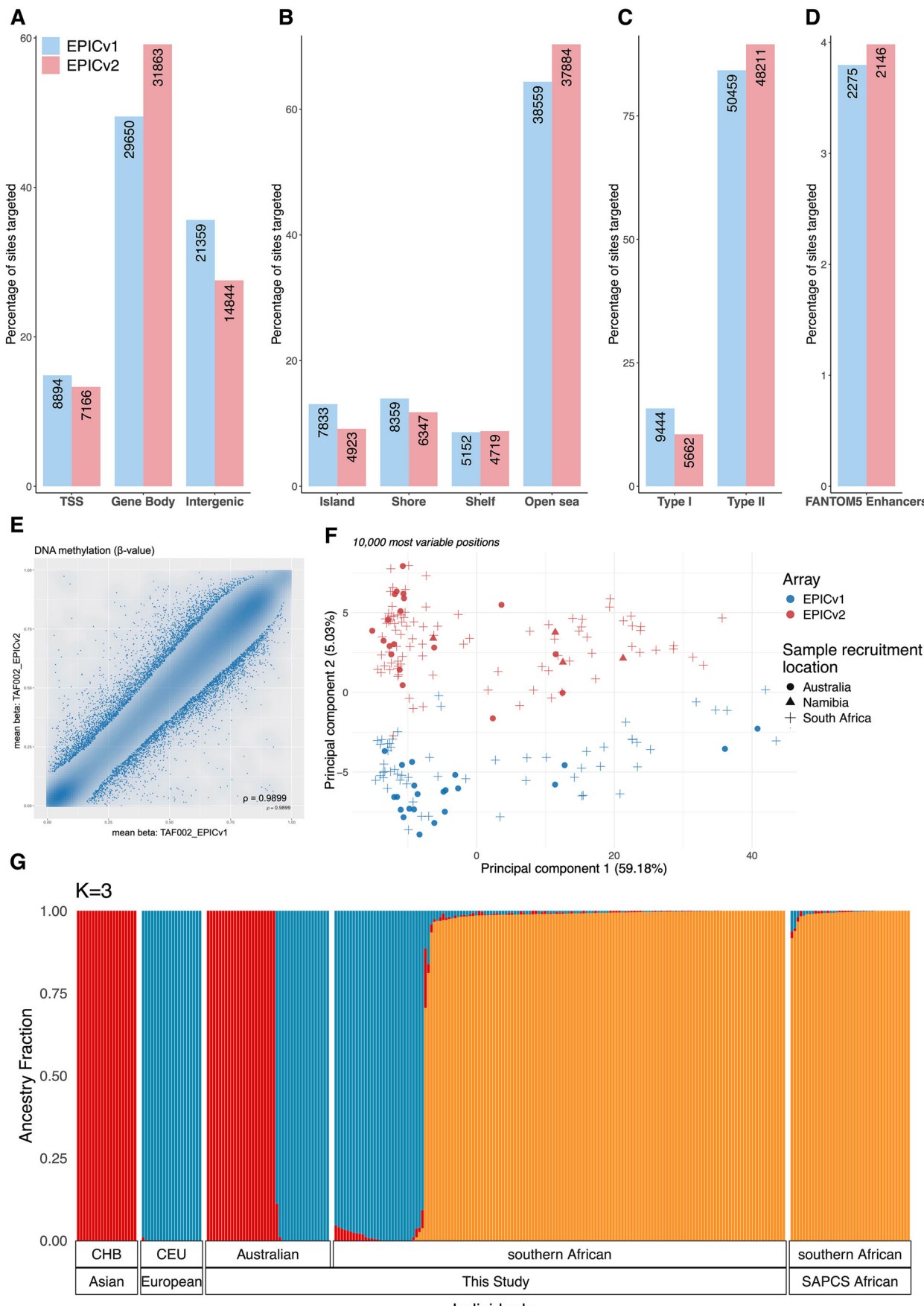

◀ **Figure 1.  DNA methylation geo-ancestral cohort analysis using the EPICv1 and EPICv2 arrays.**

(A–D) Distribution of African-confounded probe loss on EPICv1 and EPICv2 arrays across genomic features. Percentage (bars) and number (labels) of EPICv1 and EPICv2 probe sites identified to be confounded by African genomic diversity, shown relative to (A) gene regions, (B) CpG island context, (C) CpG probe type and (D) FANTOM5 enhancer overlap. Percentages are calculated relative to the total number of African-confounded probes lost per platform. (E) Scatterplot showing the correlation between methylation measurements from EPICv1 and EPICv2 for African replicate pair TAF002. (F) Principal component analysis (PCA) of a 193-sample merged and normalised EPICv1/EPICv2 dataset across the 10,000 most variable positions. Samples are annotated according to array version: EPICv1 ($n = 78$) and EPICv2 ($n = 115$); and the location from which samples were recruited: Australia ($n = 41$), Namibia ($n = 4$) and South Africa ($n = 148$). (G) Admixture plot ($K = 3$, cross-validation error = 0.524) replicated in 10/10 runs, including 40 southern Africans from the Southern African Prostate Cancer Study (SAPCS), both including and excluding for Khoe-San fractions (Jaratlerdsiri et al, 2022), and 20 Europeans (CEU) and 20 Chinese (CHB) from the Human Genome Diversity Project (HGDP) and 1000 Genomes Project (1KGP) subset of gnomAD v3.12 (Chen et al, 2023), together with our geo-ancestral cohort ($n = 192$, with exclusion of a single African with insufficient genome sequencing coverage). Source data are available online for this figure.

noise from irrelevant cell types in our African samples, only those belonging to the highest tumour purity quartile were included (reducing our initial quality-controlled study cohort from 282 to 193, see "Methods"). Consequently, ancestry is significantly confounded with certain cell type proportions, necessitating validation of ancestry-associated differential methylation findings, as described above.

## Ancestry-associated prostate tumour differential methylation discovery

The EPICv1 "ancestry-associated discovery cohort" (Table 1) was used to perform geo-ancestral differential methylation analyses between prostate tumours derived from patients of African ($n = 21$, all South African) and non-African ancestry ($n = 49$; 27 European South Africans, 22 Asian Australians). PCA revealed that despite shared geography between Black and White South Africans, tumour-driven DNA methylation grouping appears to be largely driven by ancestry, with PC1 explaining 78.76% of the total methylation variance and most strongly associating with ancestry, following tumour purity (Fig. 2A; Appendix Fig. S3; Dataset EV8). While little difference was observed between European and Asian tumours, African tumours (despite all being from the highest tumour purity quartile) exhibited a slightly broader spread along PC1 relative to non-African high-purity tumours. This observation raises the possibility of greater ancestry-associated methylation diversity, although this could also reflect residual tumour purity effects (Appendix Fig. S4). Notably, global methylation levels at repetitive elements (Alu, LINE-1, LTR) did not differ significantly by ancestry (Appendix Fig. S5).

Overall, we identified 861 differentially methylated positions (DMPs) associated with patient ancestry (African *versus* non-African BH false discovery rate (FDR) $P < 0.05$, $|\Delta\beta| \geq 20\%$) (Dataset EV9), with the top ten impacting cancer-associated genes (Table 2). As shown in Fig. 2B, these DMPs indicate ancestry-linked methylation divergence. The inclusion of African normal prostate samples further revealed that the methylation profiles at these loci were distinct from both African and non-African tumours, consistent with the tumour-specific nature of the DMPs. Compared to non-Africans, 721 DMPs (83.74%) were hypermethylated and 140 DMPs hypomethylated in Africans (Fig. 2C). Although gene expression regulation by DNA methylation is often discussed in the context of protein-coding relevant regions like CGIs and neighbouring areas (within 2 kb, i.e., CGI shore), as well as promoter regions, it is also true that enhancer elements are enriched with disease-associated epigenetic changes (Aran et al,

2013). A large proportion of ancestry-associated DMPs were located in intergenic regions (IGR) (39.37%) and open sea CpG contexts (70.38%) (Fig. 2D). Enrichment analysis using prostate tumour-derived chromatin states (Pomerantz et al, 2020) showed predominant localisation to heterochromatin (37.86%, $P = 3.15 \times 10^{-8}$, Fisher's exact test), followed by enhancers (26.71%), including active non-prostate lineage enhancers (8.48%, $P = 9.16 \times 10^{-4}$), and active and primed prostate lineage-specific enhancers (7.08%, $P = 7.58 \times 10^{-10}$ and 4.53%, $P = 3.48 \times 10^{-5}$, respectively). Significant localisation was also observed in promoters (26.25%), particularly active non-prostate lineage promoters (11.38%, $P = 8.50 \times 10^{-6}$) (Fig. 2E; Dataset EV9). Stratification by methylation direction revealed that hypermethylated DMPs were enriched in heterochromatin (41.61%, $P = 9.10 \times 10^{-13}$), whereas hypomethylated DMPs were enriched in enhancers (46.43%, $P = 1.49 \times 10^{-6}$) (Fig. 2F; Appendix Fig. S6). Notably, promoters and repressed chromatin show near equal hyper- and hypomethylation distribution (promoter 26.63% and 24.29%, repressed chromatin 8.88% and 10.71%, respectively).

Using DMRcate (Peters et al, 2015), we further identified 110 differentially methylated regions (DMRs) comprising 914 CpGs (FDR $P < 0.05$, $|\Delta\beta| \geq 10\%$), with the vast majority (91.82%) hypermethylated in African over non-African tumours (Datasets EV10 and EV11). Again, all top ten significant DMRs impact genes with oncogenic potential (Table 2), with DMRs largely overlapping promoter regions and CGIs (Fig. 2D,E). While hypermethylated DMRs predominantly overlapped CGIs and shores (44.96% and 39.11%, respectively), hypomethylated DMRs were more frequently found in non-CGI regions (61.67%) (Fig. 2F), with minimal overlap with common partially methylated domains (PMDs) (8.33%). Relative to annotated chromatin states, hypermethylated (78.45%, $P < 0.0001$) over hypomethylated (46.67%, $P = 1.75 \times 10^{-2}$) DMRs were more common to promoters, with hypomethylated DMRs showing predominant enhancer distribution (51.67%, $P = 6.60 \times 10^{-5}$). More specifically, hypomethylated DMRs were found within active promoters from non-prostate lineage-specific tissues (20%) and primed enhancers of prostate lineage origin (16.67%) (Appendix Fig. S6).

To formally evaluate methylation variability at the ancestry-associated DMPs, Levene's test for equality of variances was applied to the top 100 DMPs, comparing African and non-African groups. In all, 31 DMPs showed significantly different variance by ancestry ($P < 0.05$), of which 26 remained significant following FDR correction. Standard deviation values within each ancestry group confirmed that, for 28 of 31 DMPs (90.3%), the African group exhibited the highest intra-group variability (Fig. EV2), supporting

**Table 1. Study participant geo-ancestries and clinicopathological prostate cancer characteristics, including tissue-associated EPIC array-based features.**

| | African (n = 122) | | European | Asian |
|---|---|---|---|---|
| | PCa (n = 57)[a] | No PCa (n = 65) | PCa (n = 48) | PCa (n = 23) |
| **Country of recruitment** (n = 193)[b] | | | | |
| South Africa (n = 148) | 56 (98.2) | 64 (98.5) | 28 (58.3) | 0 |
| Namibia (n = 4) | 1 (1.8) | 1 (1.5) | 2 (4.2) | 0 |
| Australia (n = 41) | 0 | 0 | 18 (37.5) | 23 (100) |
| **Ancestral fraction** (n = 192)[c] | | | | |
| African (n = 121)[d] | 0.987 (0.040) | 0.989 (0.024) | 0 | 0 |
| Non-African (n = 71) | 0 | 0 | 0.987 (0.023) | 1 (0) |
| **Clinicopathological characteristics** | | | | |
| Age (n = 193) | | | | |
| Mean (years) | 68.5 | 67.4 | 64.7 | 67.4 |
| Range (years) | 47–99 | 51–87 | 51–78 | 51.6–77.8 |
| PSA level (n = 190)[e] | | | | |
| Mean (ng/mL) | 196.2 (542.5) | 35.5 (97.6) | 15.1 (20.7) | 8.8 (2.5) |
| Range (ng/mL) | 0.04–3428 | 1.67–761 | 3.50–128 | 5–14 |
| ISUP (n = 192)[f] | | | | |
| 1–2 (n = 62) | 30 (52.6) | NA | 26 (54.2) | 6 (26.1) |
| 3–5 (n = 65) | 27 (47.4) | NA | 21 (43.8) | 17 (73.9) |
| **EPIC array characteristics** | | | | |
| v1.0 (n = 78) | 22 (38.6) | 7 (10.8) | 27 (56.3) | 22 (95.7) |
| v2.0 (n = 115) | 35 (61.4) | 58 (89.2) | 21 (43.8) | 1 (4.3) |
| **Estimated cell type proportion** (n = 193) | | | | |
| T-luminal | 49.6 (11.2) | 6.0 (5.2) | 13.1 (15.9) | 19.2 (18.9) |
| Stromal | 23.6 (8.2) | 43.5 (10.3) | 42.3 (11.9) | 35.4 (10.5) |
| Immune | 16.4 (6.6) | 26.0 (8.1) | 19.9 (6.2) | 19.3 (7.4) |
| **Ancestry-associated cohort** | | | | |
| Discovery (v1.0, n = 70) | 21 | NA | 27 | 22 |
| Validation (v2.0, n = 57) | 35 (n = 7 presumably PCa) | NA | 21 | 1 |
| **Tumour-associated cohort** | | | | |
| Discovery (v2.0, n = 93) | 35 (n = 7 presumably PCa) | 58 | NA | NA |
| Validation (v1.0, n = 29) | 22 | 7 | NA | NA |

*ISUP* International Society of Urological Pathology, *NA* not applicable, *PCa* prostate cancer, *PSA* prostate-specific antigen

[a]Including seven No PCa patients reclassified as "presumably PCa" through methylation profiling.
[b]South African and Namibian patients are merged geographically as southern African.
[c]Ancestry fraction was unavailable for a single No PCa African patient.
[d]Including a single patient with African (70.5%) and Asian (18%) admixture.
[e]PSA was missing for two African (1 PCa, 1 No PCa) and one European South African patient.
[f]ISUP unavailable for a single Namibian European PCa patient. ISUP 1–2 includes seven African patients with no observed pathological PCa reclassified as "presumably PCa" through methylation profiling. Continuous variables are reported as mean (standard deviation), and categorical variables are reported as number (%).

greater methylation heterogeneity in African tumours. To assess whether ancestry-associated differences were confounded by tumour purity, we examined methylation levels at the top ten DMPs and DMRs (Table 2) across all tumour samples. While both DMP scatterplots and boxplots showed ancestry-associated differences independent of tumour purity (Appendix Figs. S7 and S8), similar DMR plots (Appendix Figs. S9 and S10) showed more variable separation.

Exclusively considering significant differential methylation overlap with our ancestry-associated validation cohort (Fig. 2G) for functional enrichment analysis, and further filtering to include genes that display a correlation between methylation and expression in the prostate (see "Methods"), we examined whether DMPs and DMRs were collectively enriched in biological pathways of interest presented in curated gene sets from the Molecular Signatures Database (MSigDB) (Subramanian et al, 2005). Within the MSigDB C2 collection, hypermethylated DMPs/DMRs were enriched for cancer-related gene sets, including genes down-regulated in metastatic prostate and breast cancers, and genes hypermethylated in lung cancer (Fig. 2H). Additional enrichment was observed in histone deacetylase (HDAC)-related gene sets (Dataset EV12). Hypermethylated loci also showed enrichment in oncogenic signatures (C6 collection) and immunologic gene sets (C7). Although poorly enriched, hypomethylated DMPs were likewise enriched in immune- and cancer-related gene sets, with notable inclusion of genes upregulated in breast cancer and in mammary stem cells. In addition, hallmark gene sets include genes involved in epithelial to mesenchymal transition (EMT). Few pathways remained significant at a 5% FDR threshold. Using GeneHancer's "Double Elite" list (Fishilevich et al, 2017), we further verified DMP and DMR overlap with 33 validated promoter/enhancer elements. Two or more sources of evidence suggest a high likelihood of interaction between these regions and 63 target genes collectively (Dataset EV13). Target genes of differentially methylated enhancers were notably enriched in developmental pathways, various gene sets of known transcription factor (TF) targets (*ETS2* proto-oncogene, *FOXO4* tumour suppressor), and in prostate cancer-associated pathways (Dataset EV14).

Validated DMP and DMR CpGs were further annotated to 97 genes (Fig. 2G), of which 16 were both DMP- and DMR-related. Besides 12 known PCa-associated genes (Dataset EV15), we found four genes to be of particular interest as potentially unknown African-specific PCa targets, namely *GALM*, *EVC2*, *CHSY1* and *SPDYA* (Fig. 3). To evaluate tumour purity confounding, we examined methylation levels at these genes by ancestry and tumour purity. These loci showed consistent ancestry-associated differences across purity strata, with greater divergence in high-purity tumours (Fig. EV3), supporting their biological relevance. Of the genes reported to be frequently differentially methylated between African American and European American prostate tumours (e.g., *RARB*, *TIMP3*) (Stevens et al, 2023), none were significantly differentially methylated in our cohort.

Validating our ancestry-associated differential methylation findings, we performed equivalent analyses in a tumour-derived EPICv2 "ancestry-associated validation cohort" for 35 African (all southern African) and 22 non-African (3 European southern Africans, 1 Asian- and 18 European Australians) patients, while

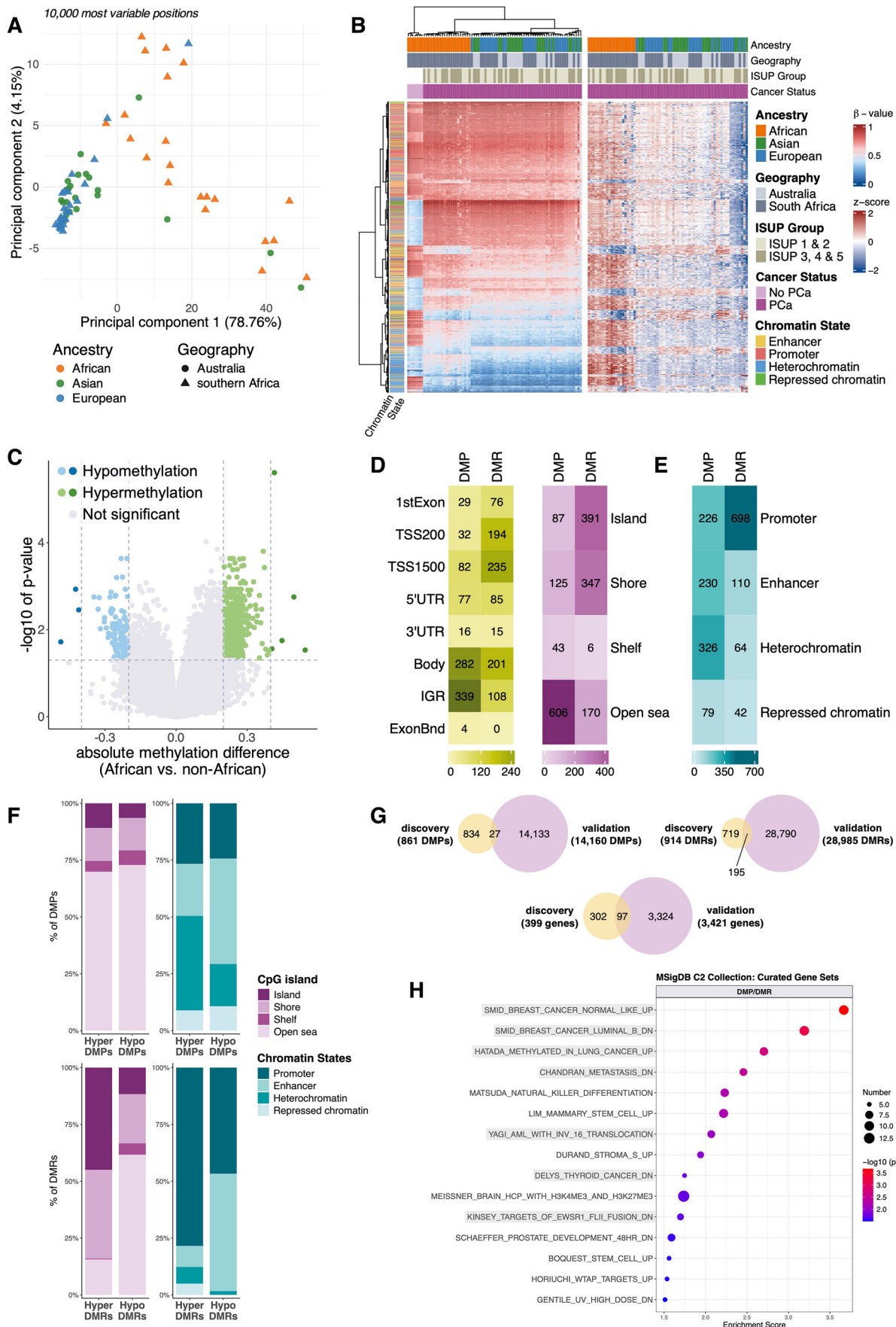

**Figure 2.   Differential DNA methylation analysis for African- *versus* non-African-derived prostate tumours.**

(**A**) PCA for the 10,000 most variable positions for the EPICv1 "ancestry-associated discovery cohort" by ancestry: African ($n = 21$), European ($n = 27$) and Asian ($n = 22$), and geography: South Africa ($n = 48$) and Australia ($n = 22$). (**B**) DNA methylation heatmaps at ancestry-associated DMPs. Left: β-values (range 0–1) for tumour ($n = 70$) and normal ($n = 7$) prostate samples. Right: Row-scaled z-scores for tumour samples only. Top annotations indicate ancestry, geography and ISUP grade group, and rows ($n = 861$ CpGs) are annotated by chromatin state. Heatmaps display absolute (left) and relative (right) methylation differences at ancestry-associated loci. (**C**) Volcano plot representing 861 DMPs by ancestral (21 African *versus* 49 non-African) significance determined using linear regression (BH FDR $p < 0.05$, $|\Delta\beta| \geq 20\%$ and 40%, dashed lines), including hypermethylated (green) or hypomethylated (blue), compared with non-significant (grey) DMPs. (**D**) Heatmap showing gene region (green) and CpG island-related region (purple) enrichment for DMP- and DMR-related CpG sites across various contexts. (**E**) Heatmap showing chromatin state enrichment of DMP- and DMR-related CpG sites across various contexts. (**F**) Stacked bar graphs of the percent overlap of DMPs and DMRs with various CpG island-related (purple) and chromatin state (blue) contexts. (**G**) Venn diagrams illustrating significant DMP, DMR (as per linear regression) and annotated gene agreement between the ancestry-associated discovery (21 African, 49 non-African) and validation (35 African, 22 non-African) cohorts. (**H**) MSigDB (C2) enrichment for validated genes collectively annotated to hypermethylated DMPs and DMRs. Enrichment analysis was performed using a hypergeometric test, with the top 15 terms (minimum 5 genes) shown. Cancer-related terms are highlighted in grey. Point size reflects the number of genes associated with each gene set. Association strength is denoted by $-\log_{10}(P$ value$)$. DMP differentially methylated position, DMR differentially methylated region, ExonBnd within 20 bases of an exon boundary, IGR intergenic region, ISUP International Society of Urological Pathology, MSigDB Molecular Signatures Database, PCA principal components analysis, TSS transcription start site, UTR untranslated region, $|\Delta\beta|$ absolute difference in mean methylation. Source data are available online for this figure.

acknowledging significant confounding between ancestry and tumour purity, minimising appropriate adjustment. Despite limited statistical overlap, 27 DMPs and 195 DMRs reached significance in the validation cohort. Of these, 100% of validated DMPs and 96.4% of validated DMRs displayed consistent direction of methylation change between discovery and validation datasets. Permutation-based enrichment tests confirmed that these overlaps were significantly greater than expected by chance ($P = 0.0023$ for DMPs, $P < 0.0001$ for DMRs), supporting non-random reproducibility. Full validation cohort results are presented in Appendix Figs. S11–S14 and Datasets EV16–EV19.

## African prostate tumour-associated differential methylation discovery

Using the African-specific EPICv2 prostate tissue-derived "tumour-associated discovery cohort", including 93 southern Africans either with ($n = 35$, including 7 "presumably PCa") or without ($n = 58$) clinicopathologically confirmed PCa (Table 1), distinguished patients by DNA methylation profile through PCA visualisation, with the spread in PC2 attributable to non-tumour stromal and immune cell content (Fig. 4A; Appendix Figs. S15 and S16; Dataset EV20). Furthermore, we found prostate tumours to be associated with increased genome-wide methylation in CpG dense regions, while conversely associated with decreased genome-wide methylation at repetitive elements (LINE-1, LTR, Alu) (Fig. 4B; Appendix Fig. S17), with the latter showing 27.50% collective overlap with common PMDs.

Through differential methylation analysis, we identified 9501 tumour-associated DMPs (BH FDR $P < 0.05$, $|\Delta\beta| \geq 30\%$) (Dataset EV21), with hypermethylation dominance (66.64%) in tumour-positive tissue (Fig. 4C), and DMP cluster analysis showing a clear distinction between tumour and non-tumour samples, with bias towards tumour-associated hypermethylation (Fig. 4D). Similar to our ancestry-associated findings, tumour-associated DMPs were primarily located in non-CGI regions (72.21%), followed by CGIs and shores (21.76%). These DMPs showed predominant enrichment in heterochromatin (34.12%, $P = 2.19 \times 10^{-23}$), followed by promoters (30.88%), including active non-prostate (12.16%, $P = 7.5 \times 10^{-61}$) and bivalent poised promoters (16.51%, $P < 0.0001$), and active non-prostate enhancers (24.26%, $P < 0.0001$) (Fig. 4E,F). To assess potential functional importance,

we determined the distribution of DMPs across various genomic contexts (Fig. 4G; Appendix Fig. S18). Consistent with our global analysis, CGI regions exhibit greater gain of methylation over loss. However, most hypo- and hypermethylated DMPs reside in non-CGI regions (90.47% and 63.07%, respectively). Relative to annotated chromatin states, hypermethylated DMPs were primarily found in promoters (45.59%, $P < 0.0001$) and enhancers (32.59%, $P = 5.16 \times 10^{-13}$), whereas hypomethylated DMPs were mostly located in heterochromatin (65.05%, $P < 0.0001$) and repressed chromatin (25.80%, $P < 0.0001$).

Furthermore, we identified 707 tumour-associated DMRs comprising 5,444 CpGs (FDR $P < 0.05$, $|\Delta\beta| \geq 20\%$) (Datasets EV22 and EV23). Compared to non-tumour samples, 423 DMRs are hypermethylated and 284 hypomethylated in the African tumours, with overall distribution in promoter regions (47.22%) (Fig. 4E,F). Consistent with global CpG methylation, hypermethylated DMRs were biased to CGI regions (49.80%), while hypomethylated DMRs predominantly occurred in open sea regions (91.54%) (Fig. 4G). With respect to chromatin states, 70% ($P < 0.0001$) of hypermethylated compared to 1.42% of hypomethylated DMRs mapped to promoters, with most hypomethylated DMRs distributed in heterochromatin (52.24%, $P < 0.0001$) and repressed chromatin (37.45%, $P < 0.0001$). Interestingly, both hypomethylated DMPs and DMRs show negligible promoter distribution (1.51% and 1.42%, respectively).

To consider African-specific differential methylation, we first performed differential methylation analysis between The Cancer Genome Atlas (TCGA)-PRAD European tumour and normal samples (see "Methods") (57,594 DMPs, BH FDR $P < 0.05$, $|\Delta\beta| \geq 10\%$, Dataset EV24), and then conducted functional enrichment analysis on 405 African tumour-associated differentially methylated CpGs, selected for their overlap with our tumour-associated validation cohort but exclusion from TCGA-PRAD cohort (Fig. 4H; Dataset EV25). Again, only annotated genes with a prostate tissue methylation/expression correlation were included. Notably, among the overlapping loci between our African-derived DMPs and TCGA DMPs, 99.98% demonstrated consistent direction of methylation change, with strong correlation in $\Delta\beta$ values ($r = 0.86$, $P < 2.2 \times 10^{-16}$), further supporting the robustness of tumour-associated methylation differences. Hypermethylated DMPs/DMRs were enriched for cancer-related gene sets (MSigDB C2 collection) defined as downregulated in (metastatic) PCa,

**Table 2.  Top ten significant differentially methylated positions (DMPs) and regions (DMRs) identified between African and non-African prostate tumours.**

| DMP | | | | | | | | | | | | DMR | | | | | | |
| --- | --- | --- | --- | --- | --- | --- | --- | --- | --- | --- | --- | --- | --- | --- | --- | --- | --- | --- |
| DMP | Δβ | African mean β | Non-African mean β | $P_{adj}$ | Chr | CpG coord | Gene | Distance to TSS of (nearest gene) | Region | Location | Chromatin state | DMR | Chr | Start | End | Δβ | $P_{adj}$ | Gene(s) |
| cg10025443 | −0.2565 | 0.1156 | −0.1409 | $2.5 \times 10^{-6}$ | 9 | 93,564,339 | SYK | 94 | Body | Island | Promoter | DMR_1 | 13 | 36,048,892 | 36,051,749 | 0.2117 | $9.2 \times 10^{-44}$ | NBEA, MAB21L1 |
| cg12498690 | 0.4156 | 0.5131 | 0.9287 | $2.5 \times 10^{-6}$ | 12 | 3,164,436 | / | 13,832 (RP11-253E3.3) | IGR | Open sea | Enhancer | DMR_2 | 2 | 42,793,873 | 42,795,624 | 0.1579 | $6.2 \times 10^{-29}$ | MTA3 |
| cg27019651 | 0.3683 | 0.2879 | 0.6562 | $1.6 \times 10^{-4}$ | 18 | 18,743,966 | / | 52,153 (ROCK1) | IGR | Open sea | Heterochromatin | DMR_3 | 6 | 125,283,726 | 125,284,659 | 0.1759 | $1.2 \times 10^{-24}$ | RNF217, RP11-510H23.1 |
| cg18576923 | 0.2355 | 0.9375 | 1.1729 | $2.3 \times 10^{-4}$ | 14 | 68,157,195 | RDH11 | 110 | Body | Open sea | Enhancer | DMR_4 | X | 78,622,433 | 78,624,108 | 0.2615 | $7.0 \times 10^{-24}$ | ITM2A |
| cg05938267 | 0.2836 | 0.8747 | 1.1584 | $2.3 \times 10^{-4}$ | 16 | 77,228,897 | MON1B | 3799 | Body | Shore | Promoter | DMR_5 | 10 | 102,414,426 | 102,415,991 | 0.1343 | $8.8 \times 10^{-24}$ | NA |
| cg04406863 | −0.2072 | 0.3933 | 0.1861 | $2.3 \times 10^{-4}$ | 20 | 62,177,351 | SRMS | 1504 | Body | Shore | Enhancer | DMR_6 | 7 | 93,519,473 | 93,520,566 | 0.1143 | $1.9 \times 10^{-23}$ | GNGT1, AC002076.10, TFPI2 |
| cg23466144 | −0.2326 | 0.8616 | 0.6290 | $2.3 \times 10^{-4}$ | 2 | 178,562,071 | PDE11A | 13,872 | Exon Bnd | Open sea | Heterochromatin | DMR_7 | 15 | 101,728,234 | 101,729,174 | 0.1606 | $1.4 \times 10^{-22}$ | CHSY1 |
| cg01486146 | 0.2415 | 0.1244 | 0.3659 | $2.3 \times 10^{-4}$ | 10 | 102,415,311 | / | 80,047 (PAX2) | IGR | Island | Promoter | DMR_8 | 16 | 77,227,670 | 77,228,897 | 0.2529 | $4.2 \times 10^{-22}$ | MON1B |
| cg12189097 | 0.2270 | 0.9217 | 1.1487 | $2.3 \times 10^{-4}$ | 1 | 207,178,154 | / | 27,936 (C1orf116) | IGR | Open sea | Enhancer | DMR_9 | 3 | 237,955 | 239,036 | 0.1426 | $6.9 \times 10^{-21}$ | CHL1, CHL1-AS2 |
| cg01459033 | 0.2685 | 0.9259 | 1.1944 | $2.4 \times 10^{-4}$ | 16 | 77,228,497 | MON1B | 3399 | Body | Island | Promoter | DMR_10 | 5 | 42,994,709 | 42,995,597 | 0.2443 | $6.4 \times 10^{-20}$ | NA |

Chr chromosome, CpG Coord chromosomal coordinates of the target CpG, DMP differentially methylated position, DMR differentially methylated region, ExonBnd within 20 bases of an exon boundary (i.e, the start or end of an exon), IGR intergenic region, NA not applicable, $P_{adj}$ BH-adjusted P value, TSS transcription start site, β beta methylation value, Δβ difference in mean methylation.

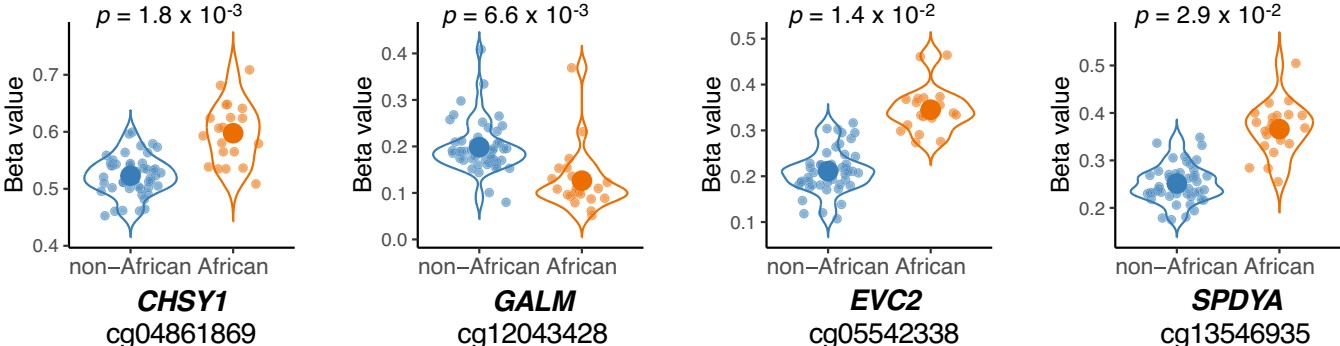

**Figure 3. African *versus* non-African CpG DNA methylation plots for four individual potentially unknown African-specific prostate cancer (PCa) targets.**

Noteworthy significant differentially methylated positions (DMPs) between African (orange, $n = 21$) and non-African (blue, $n = 49$) PCa samples. The $y$ axis represents sample beta values for each individual CpG probe, after covariate adjustment. The FDR significance in the difference of mean beta values between the two groups, as per a $t$ test, is shown. Source data are available online for this figure.

downregulated upon HDAC knockdown (Fig. 4I) and silenced by methylation in several cancer types (Dataset EV26). Hallmark gene sets included TNF-α signalling and cholesterol homeostasis. Conversely, hypomethylated DMPs/DMRs exhibited sparse gene set enrichment, unsurprising given 50% of the hypomethylated CpGs fall within a common PMD. Not all pathways were significant after controlling for a 5% FDR. Utilising the GeneHancer 'Double Elite' list, we verified DMP and DMR overlap with 396 validated promoter/enhancer elements. Two or more sources of evidence suggest a high likelihood of interaction between these regions and 749 target genes collectively (Dataset EV27), with these genes showing enrichment in gene sets of known TF targets and numerous cancer-associated pathways (Dataset EV28). Hypermethylated elements included target genes involved in tumour-suppressor binding motifs (*FOXO4*, *SMAD2*, *TCF21*, *RB1*, *TP53*), whereas target genes of hypomethylated elements belong to developmental pathways and oncoprotein binding motifs, including E12 (encoded by *TCF3*, a PCa oncogene).

Of the 405 CpGs found to be African-specifically differentially methylated in this study, 27 CpGs displayed a DMP/DMR $|\Delta\beta| \geq 20\%$, annotated to 16 genes: *GJB5*, *CD1E*, *C2orf88*, *SLC19A3*, *PROM1*, *ARL9*, *MIR575*, *SLC12A9*, *LRRC4*, *TACC1*, *HOXC4*, *ADCY4*, *PYCARD*, *CX3CL1*, *FBXO17* and *KLF8*. While 11 genes, including *PYCARD*, have PCa associations (Dataset EV29), it appears that *GJB5*, *CD1E*, *SLC19A3*, *SLC12A9* and *FBXO17* have not been previously reported in relation to PCa in published studies. However, we found two CpGs to be of particular interest as potentially unknown African-specific PCa CpG targets: cg04742719 (*SLC12A9*) and cg11970458 (*PYCARD*) (Fig. 5). We also observed tumour-associated hypermethylation amongst several of the most extensively studied and independently validated DNA methylation biomarkers for PCa (Lam et al, 2020), including *GSTP1*, *CCND2* and *PDLIM4*.

Equivalent analyses were performed using our independent "tumour-associated validation cohort", including EPICv1 prostate tissue-derived data from southern African men either with ($n = 22$) or without ($n = 7$) clinically confirmed PCa, again, acknowledging confounding in this cohort between our variable of interest and

stromal cell content. Validation cohort results can be found in Appendix Figs. S19–S23 and Dataset EV30–EV33.

## Discussion

The inclusion of African populations across the rich geographic diaspora is crucial for uncovering the genetic and environmental factors that contribute to variation in PCa risk and associated clinical adversity. This is further enhanced through the study of epigenetic modifications, with a focus on DNA methylation, a common feature of tumorigenesis and associated environmental exposures. However, epigenomic studies involving men of African descent remain predominantly limited to African American cohorts (Samtal et al, 2022), with a notable lack of data for the African continent. Representing extreme genetic and environmental diversity (Choudhury et al, 2017), as well as the greatest global PCa mortality rates (Bray et al, 2024), here we focus on southern Africa. Establishing a regionally relevant filtering resource for Illumina EPIC-generated DNA methylation datasets, we provide characterisation of ancestry-associated methylation differences and African tumour-associated epigenetic alterations in prostate tissue. Our findings highlight ancestry-over geographic-associated prostate tumour methylation, suggest potentially greater African-specific heterogeneity and point towards ancestry-informed biomarkers.

Given the shared evolutionary history (Out of Africa) and in turn, genetic similarity between Asian and European individuals, both distant from Africans (Nei and Livshits, 1989), it appears genomic characterisation by ancestry, including in our study multi-generational European southern Africans, is mirrored in the PCa methylome. Overall, we observed that both ancestry- and tumour-associated methylation changes were predominantly characterised by hypermethylation, particularly at promoters and CGIs, consistent with canonical epigenetic silencing mechanisms (Skvortsova et al, 2019). In African tumours, we observed greater hypermethylation of DMRs in protein-coding regions, with hypermethylated loci enriched in regions associated with tumour-suppressor regulation and metastatic disease. This suggests that epigenetic repression of tumour-suppressor genes may be more pervasive or

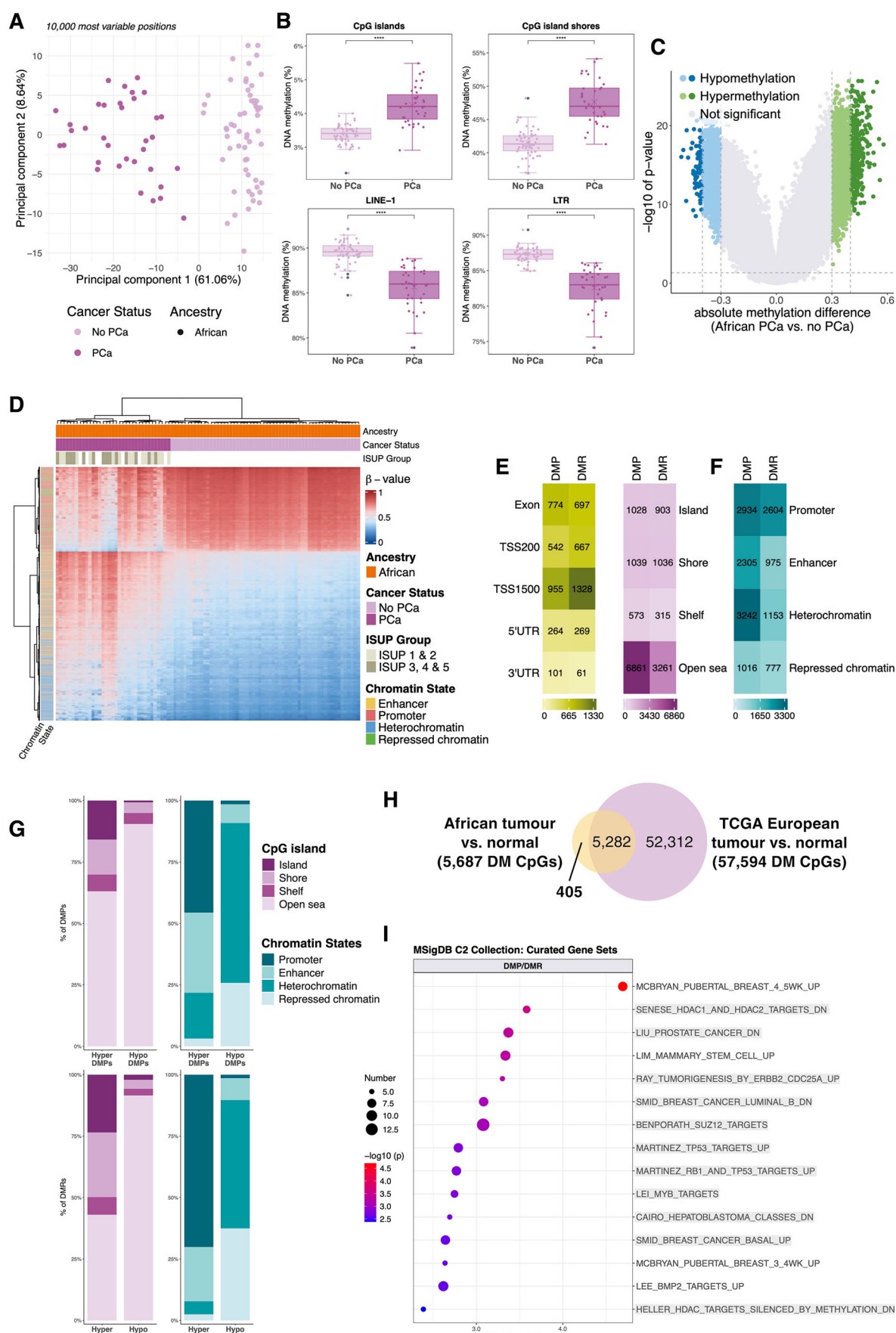

**Figure 4. African-specific prostate tumour *versus* normal tissue differential methylation.**

(A) PCA for the 10,000 most variable positions for the EPICv2 African "tumour-associated discovery cohort" by cancer status: those with PCa (*n* = 35), and those without PCa (*n* = 58). (B) Boxplots of global DNA methylation levels in African prostate tumour *versus* normal tissue samples across CpG islands, CpG island shores, LINE-1 repetitive elements and LTRs. The box represents the interquartile range (IQR), with the bottom and top boundaries indicating the 25th and 75th percentiles, respectively. The horizontal line within the box shows the median (50th percentile). The whiskers extend from the box to the minimum and maximum values within 1.5 times the IQR, with individual points beyond the whiskers considered outliers. Each dot represents the median methylation value for each patient. The "X" indicates group mean. (C) Volcano plot representing 9501 DMPs by cancer status (35 prostate tumour *versus* 58 normal tissue) significance determined using linear regression (BH FDR *P* < 0.05, |Δβ| ≥ 30% and 40%, dashed lines), including hypermethylated (green) or hypomethylated (blue), compared with non-significant (grey) DMPs. (D) DMP cluster analysis heatmap by cancer status, ISUP grade group and chromatin state. Rows represent CpG sites (*n* = 9501) and columns, patients (*n* = 93). The methylation level is represented by a β-value between 0 (completely unmethylated, blue) and 1 (fully methylated, red). (E) Heatmap showing gene region (green) and CpG island-related region (purple) enrichment for DMP- and DMR-related CpG sites across various contexts. (F) Heatmap showing chromatin state enrichment of DMP- and DMR-related CpG sites across various contexts. (G) Stacked bar graphs of the percent overlap of DMPs and DMRs with various CpG island-related (purple) and chromatin state (blue) contexts. (H) Venn diagram illustrating shared and distinct significantly differentially methylated CpGs (BH FDR *P* < 0.05, |Δβ| ≥ 10%) identified in the African tumour-associated cohorts and TCGA-PRAD European cohort. (I) MSigDB (C2) enrichment for genes collectively annotated to hypermethylated DMPs and DMRs not identified within TCGA-PRAD European cohort. Enrichment analysis was performed using a hypergeometric test, with the top 15 terms (minimum 5 genes) shown. Cancer-related terms are highlighted in grey. Point size reflects the number of genes associated with each gene set. Association strength is denoted by −log10(*P* value). (A–I) For all tumour-associated analyses, the 35 African tumours include 7 samples reclassified as "presumably PCa" through methylation profiling. DM differentially methylated, DMP differentially methylated position, DMR differentially methylated region, ISUP International Society of Urological Pathology, LTRs long tandem repeats, MSigDB Molecular Signatures Database, PCA principal component analysis, TCGA The Cancer Genome Atlas, TSS transcription start site, UTR untranslated region, |Δβ| absolute difference in mean methylation; ****P* ≤ 0.0001. Source data are available online for this figure.

target additional loci in African-derived tumours, potentially contributing to aggressive phenotypes. This is supported by enrichment for gene sets downregulated in metastatic PCa (Chandran et al, 2007).

Tumour-associated hypermethylation also displayed substantial enhancer distribution. As enhancers regulate gene expression via TF recruitment and DNA looping to promoters, their aberrant methylation can significantly alter (cancer-associated) transcriptional programmes (Aran et al, 2013). We again observed enrichment of enhancer target genes among those downregulated in metastatic PCa, indicating both direct transcriptional repression through aberrant methylation and indirect effects via epigenetically altered interacting enhancers. Ancestry-associated enhancer targets included genes regulated by the *ETS2* proto-oncogene and *FOXO4* tumour suppressor, while tumour-associated hypermethylated promoter/enhancer elements corresponded to genes harbouring binding motifs for key tumour suppressors (FOXO4, SMAD2, TCF21, RB1, TP53). FOXO4 suppresses metastasis by inactivating the oncogenic PI3K/AKT pathway (Su et al, 2014). Given DNA methylation represses enhancer activity (Song et al, 2019), reduced tumour-suppressor function in African PCa may occur through methylation of enhancer binding regions. This suggests an oncogenic mechanism whereby enhancer hypermethylation disrupts TF recruitment, binding dynamics and enhancer-promoter looping, ultimately altering target gene expression.

In contrast, ancestry-associated hypomethylation was predominantly enriched in enhancers and promoters, potentially reflecting the activation of latent oncogenic pathways. These regions were enriched for (Skvortsova et al, 2019) EMT, immune signalling targets and TF binding motifs involved in oncogenic signalling, including ETS2, E2F5 and E2F2. Notably, ancestry-associated aberrant methylation also showed predominant enrichment in heterochromatin, suggesting disruption of chromatin structure maintenance and consequent genomic instability (El-Osta, 2004). Similarly, tumour-associated hypomethylated regions were enriched in repressed chromatin and heterochromatin, indicating altered chromatin accessibility and relaxation of epigenetic constraints (El-Osta, 2004) in African tumours. This is consistent

with prior reports of increased genomic instability in African PCa (Gong et al, 2022). Supporting this, both ancestry- and tumour-associated hypermethylation showed enrichment for gene sets indicative of reduced HDAC activity (Heller et al, 2008; Senese et al, 2007), aligning with previously reported African-specific epigenetic machinery alteration of HDACs (Craddock et al, 2023). Given that transcriptionally inactive heterochromatin is maintained by HDAC-mediated histone deacetylation, these findings collectively point to a relaxation of chromatin conformation.

Our analyses further revealed a shared theme across ancestry- and tumour-associated methylation profiles, in which both were disproportionately enriched in regulatory elements linked to developmental lineages outside the prostate. Ancestry-associated hypomethylation was enriched in active non-prostate lineage enhancers and promoters, and more prominent at primed enhancers, possibly reflecting activation of lineage-inappropriate developmental programmes and contributing to greater molecular heterogeneity in African tumours. In contrast, tumour-associated hypermethylation preferentially affected bivalent promoters, regions that maintain genes in a poised but repressed state, suggesting transcriptional silencing of developmental regulators during tumour progression. These patterns imply that ancestry may predispose to broader regulatory plasticity through enhancer reprogramming, while tumour evolution reinforces developmental regulatory repression via promoter hypermethylation. This is consistent with prior evidence that prostate tumours frequently re-engage developmental circuits (Pomerantz et al, 2020), and raises the possibility that ancestry modulates the extent or nature of this activation. Notably, while DMP/DMR-associated genes better indicated immune pathway alterations, enhancer target genes showed greater enrichment for developmental pathways, underscoring enhancers as key mediators of African-specific developmental dysregulation. We hypothesise that heterochromatin hypomethylation in African tumours, combined with enhancer reprogramming and diminished developmental regulation, creates a more permissive chromatin state, enabling aberrant activation of developmental genes, thereby promoting aggressive tumour

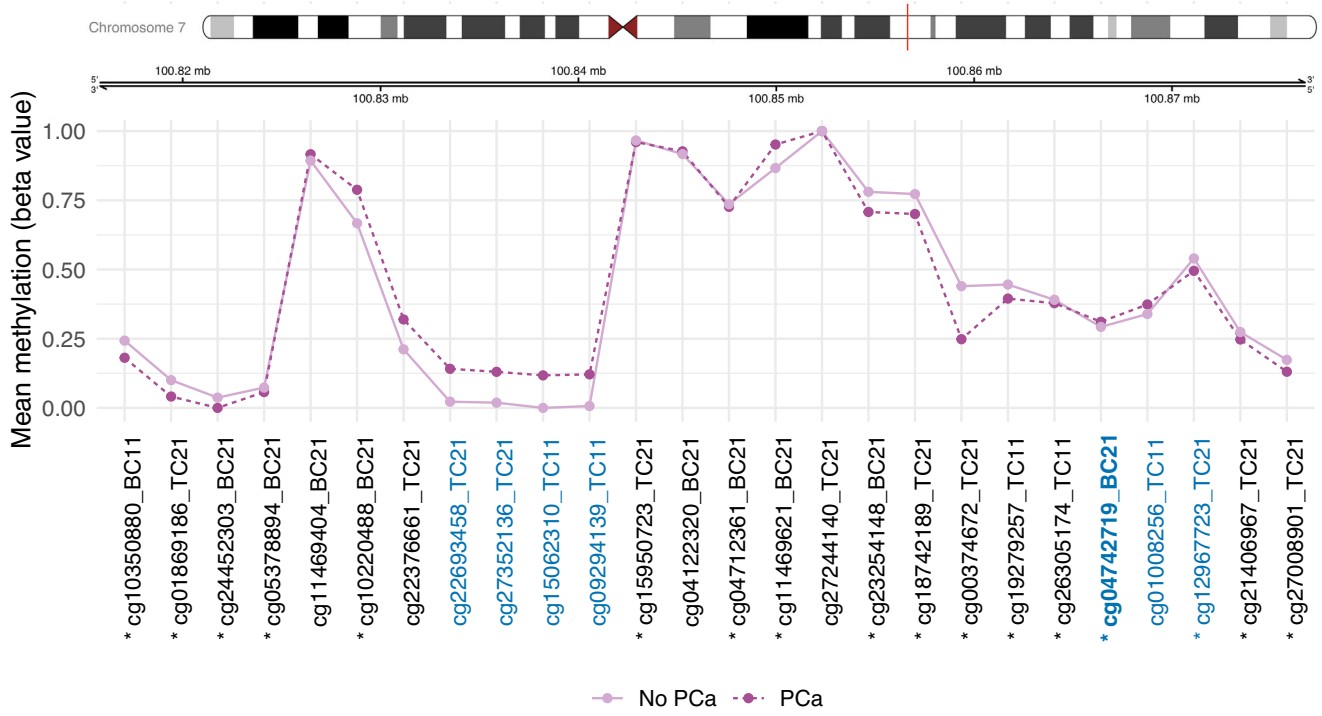

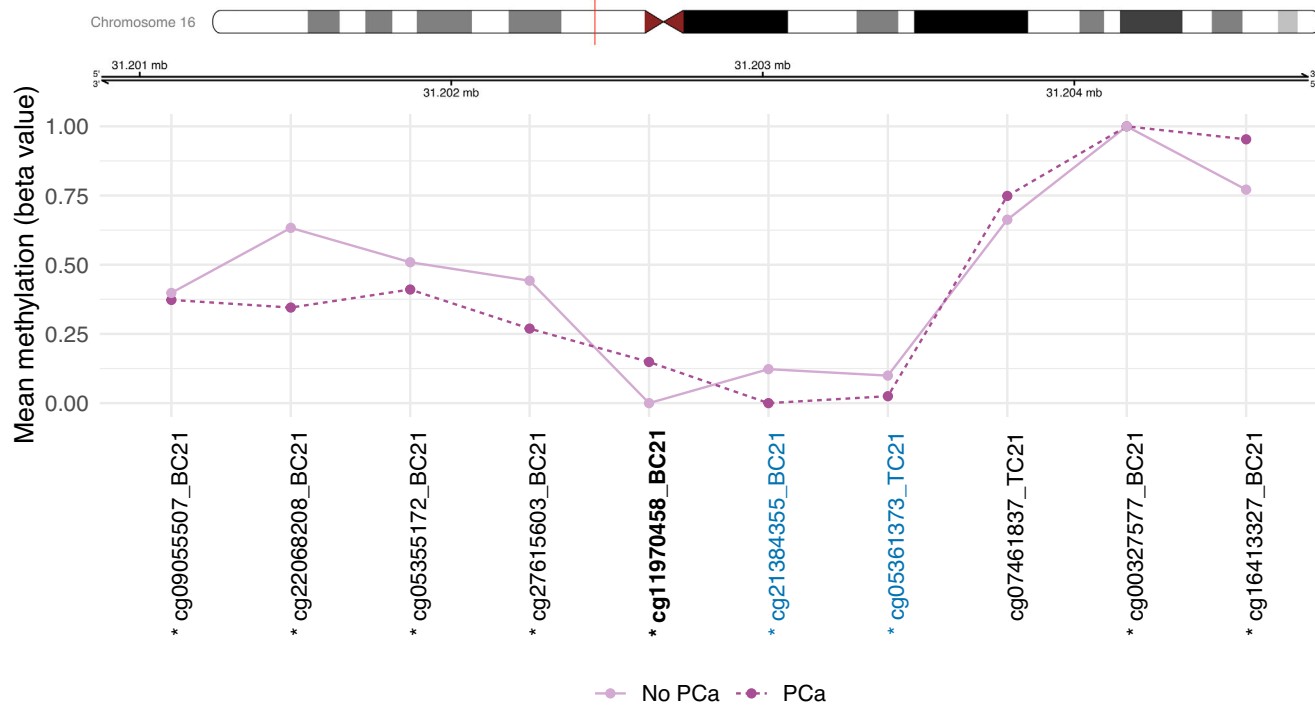

**Figure 5.   African prostate tumour *versus* normal tissue gene methylation plots highlighting two individual potentially unknown African-specific CpG prostate cancer targets.**

Differential methylation over noteworthy genes *SLC12A9* and *PYCARD*, highlighting (in bold) significant potentially unknown African-specific prostate tumour *versus* normal CpG targets. Probes overlapping CpG islands are shown in blue, the *y* axis represents the mean beta value per group for each individual CpG probe, after covariate adjustment, and differential mean methylation significance between groups, determined by FDR $P < 0.05$, is represented by an asterisk (*). Source data are available online for this figure.

phenotypes. Collectively, these findings suggest that inherited and acquired epigenetic changes disrupt lineage fidelity through distinct yet partially converging mechanisms, ultimately contributing to developmental reprogramming in African PCa.

The top hit ancestry-associated DMPs/DMRs impact genes involved in cell adhesion and neuronal development (*NBEA*, *CHL1*), development and cell differentiation (*MAB21L1*, *RDH11*), immune signalling (*SYK*), protein degradation (*RNF217*) and angiogenesis inhibition (*TFPI2*). Hypermethylated in African tumours, *NBEA* has been shown to be downregulated in PCa (Schaeffer et al, 2008), *CHL1* implicated in PCa predisposition (Rökman et al, 2005) and *RNF217* has shown reduced copy number/expression in radioresistant LNCaP (Seifert et al, 2019). However, among the most compelling findings are unknown ancestry- or tumour-associated African-specific PCa target genes, which include functionality ranging from developmental signalling (*EVC2*, *CHSY1*), cell cycle regulation (*SPDYA*), ion homeostasis (*SLC12A9*), glucose metabolism (*GALM*), to inflammatory response and apoptosis (*PYCARD*). *CHSY1* shows gene body hypermethylation (positively correlated with expression) in African tumours and correlates with poor prognosis in gastric cancer (Liu et al, 2021), while accumulation of its chondroitin sulfate chains is linked to prostate tumour aggressivity (Tóth et al, 2022), although no research has reported PCa gene-specific alterations. Hypomethylated in African tumours, *GALM* over-expression is associated with poor prognosis in glioma (Xu et al, 2022). While *SPDYA* (also *Spy1*) typically shows overexpression in ovarian cancer (Lu et al, 2016), its hypermethylation in African tumours suggests overexpression may be a mechanism unique to non-African tumorigenesis. Another notable observation includes *EVC2*'s involvement in the cancer-implicated Hedgehog signalling pathway (Jing et al, 2023), though its specific role in PCa requires further investigation. While *PYCARD* (cg11970458), but not *SLC12A9* (cg04742719), exhibits a role in PCa, the observed tumour *versus* normal hypermethylation of these particular CpGs appears to be African-specific. Upregulated solute carrier *SLC12A9* is associated with aggressive uveal melanoma (Yan et al, 2023), while *PYCARD* encodes an apoptosis-inducing factor and is characteristically suppressed by hypermethylation in aggressive PCa (Miyauchi et al, 2021). Notably, no ancestry-related differentially methylated genes typically reported in African American PCa overlapped with our southern African cohort, likely reflecting ancestral genetic divergence (Choudhury et al, 2017), reiterating the need for further inclusion across the broad African diaspora.

Our study highlights challenges associated with DNA methylation studies across Africa. Observing substantially more African *versus* European variant (Pidsley et al, 2016) overlap with array probes, although notably reduced for the updated EPICv2, raises the need for population-tailored approaches across Sub-Saharan Africa to minimise false positives due to genetic confounding. Further studies should explore applicability in larger, unrelated cohorts, for concordance between pan-African genomic variation (Fan et al, 2019). While our African ancestral prostate tumours showed extensive between tumour DNA methylation heterogeneity, seven of 72 (9.72%) non-cancer pathology-defined biopsies resembled tumour-specific methylation. Assuming misclassification, it is notable that African American men are more likely to

present with an anterior presenting tumour (Sundi et al, 2014), which is harder to detect using digitally guided transrectal prostate biopsy, the standard diagnostic approach for our southern African patients. Besides misclassification, this may also result in reduced tumour purity, possibly accounting for the heightened cell type heterogeneity observed. As such, our findings must be considered within the inherent challenges of disentangling biological variability from tumour purity confounding. While we adjusted for tumour purity using both filtering and model-based approaches in our ancestry-associated analyses, some residual confounding from cell content (including immune and stromal, though minimal) may persist. We further acknowledge the limited number of high-purity non-African samples in the ancestry-associated analyses. Moreover, while EPIC arrays have substantial overlap, differences in probe coverage limit array version validation to shared probes, potentially missing significant findings unique to each. Our analysis is further limited by African-specific CpG target identification using available TCGA 450k array European data, losing valuable information, while lack of patient-matched RNA and ChiP-seq data, limits for direct correlation between methylation changes and gene expression. The latter would be beneficial to confirm tissue-specific enhancer and chromatin state (in)activation. Additionally, the absence of matched tumour genomic data, such as *TMPRSS2-ERG* fusion status, although reportedly less frequent in southern African-derived tumours (Blackburn et al, 2019), precluded relevant adjustments. Finally, while disentangling ancestry from environmental exposures was beyond the scope of this study cohort, it remains crucial, and future studies should integrate data from genetically matched individuals across geographic regions to isolate these effects.

In conclusion, African prostate tumours exhibited distinct methylation patterns akin to aggressive disease, characterised by three key features (Fig. 6): (i) widespread hypermethylation silencing metastasis-related and tumour-suppressor genes (Fig. 6A); (ii) aberrant methylation at distal enhancers disrupting TF interactions, particularly at motifs linked to metastasis and tumour suppression (Fig. 6B); and (iii) global hypomethylation of heterochromatic regions (Fig. 6C), promoting developmental gene activation (facilitated by enhancer activation) and influencing cell identity. Overlapping features between ancestry- and tumour-associated analyses suggest tumorigenesis as a key driver of African-specific methylation, with preferential exploitation of regulatory elements linked to non-prostate developmental lineages. Ancestry may influence the extent and nature of lineage-inappropriate programme activation via enhancer hypomethylation, while tumour progression reinforces repression of developmental regulators through promoter hypermethylation—both likely contributing to aggressive phenotypes. The concurrent gain and loss of methylation at intergenic regulatory elements, along with activation of typically repressed chromatin, highlights the extensive epigenetic plasticity and resulting genomic instability in African tumorigenesis. These findings emphasise the importance of analysing regulatory regions beyond promoters, highlighting that African-specific tumorigenesis emerges from complex genome-wide epigenetic interactions rather than isolated gene-level changes. As such, we demonstrate how African diversity holds rich potential to uncover unknown insights.

   

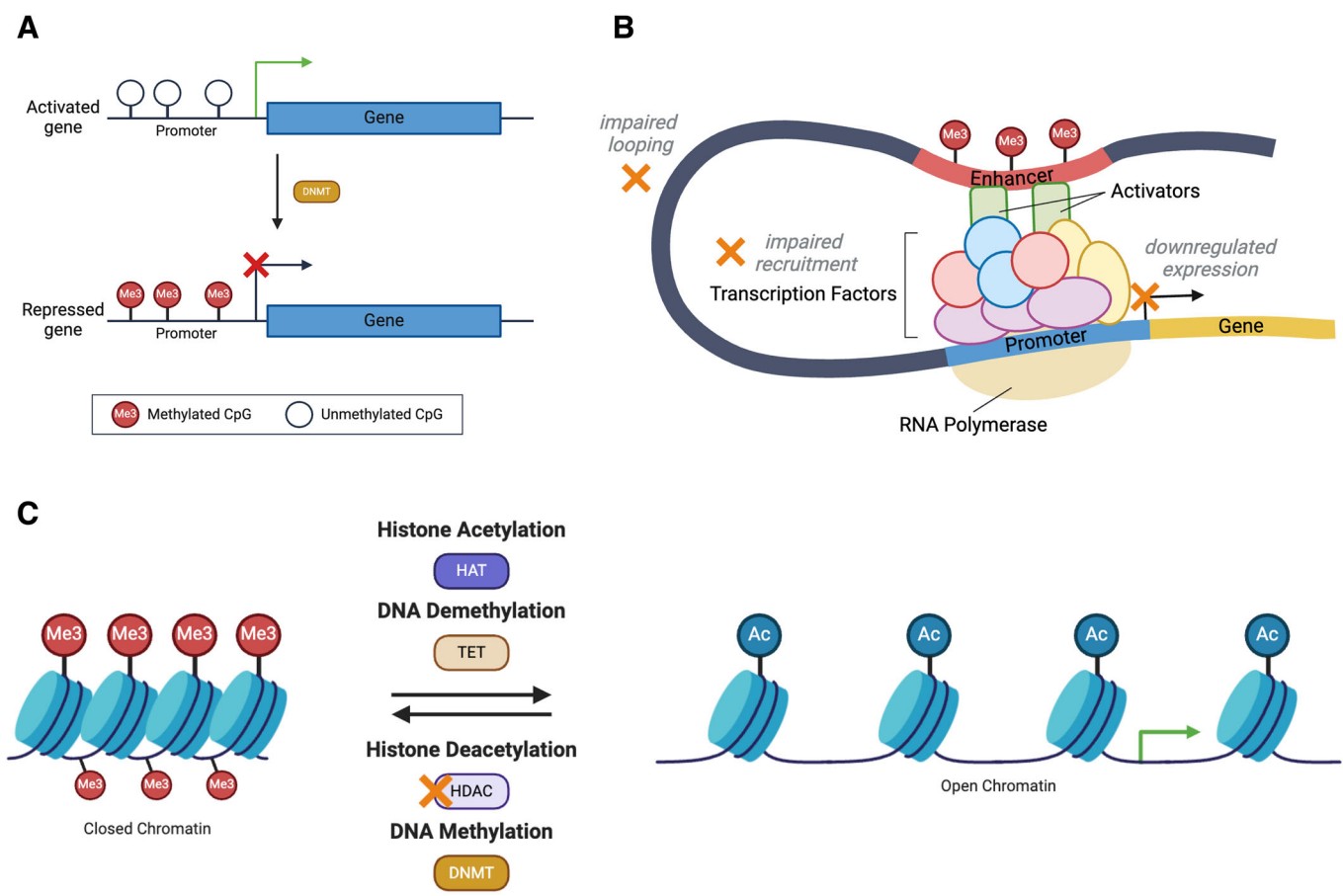

**Figure 6. Proposed model of DNA methylation-driven epigenetic aberrations in African prostate tumours.**

(A) Promoter hypermethylation-induced epigenetic silencing of (tumour-suppressor) genes. (B) Enhancer hypermethylation disrupts transcription factor (e.g., FOXO4) recruitment and binding dynamics, and impairs enhancer-promoter looping, ultimately altering target gene expression. Target genes of aberrantly methylated enhancers were often involved in tumour-suppressor binding motifs. (C) DNA hypomethylation of closed chromatin promotes a more permissive chromatin state, which may be aided by diminished HDAC activity, collectively contributing to genomic instability. DNMT DNA methyltransferase, HAT histone acetyltransferase, HDAC histone deacetylase, TET ten-eleven translocation. Created in BioRender. https://BioRender.com/rx2ybm9.

## Methods

**Reagents and tools table**

| Reagent/resource | Reference or source | Identifier or catalogue number |
|---|---|---|
| **Experimental models** | | |
| *Homo sapiens* | This study (variant data) | PRJEB94535 |
| *Homo sapiens* | This study (methylation data) | GSE304293 |
| **Recombinant DNA** | | |
| **Antibodies** | | |
| **Oligonucleotides and other sequence-based reagents** | | |
| **Chemicals, enzymes and other reagents** | | |
| **Software** | | |
| R software v4.4.1 | https://www.r-project.org/ | |

| Reagent/resource | Reference or source | Identifier or catalogue number |
|---|---|---|
| Picard v2.21.9 | Van der Auwera and O'Connor, 2020 | |
| bcftools v1.9 | Danecek et al, 2021 | |
| **Other** | | |
| QIAamp DNA Mini Kit (50) | Qiagen | 51104 |
| Illumina Infinium HumanMethylationEPIC v1.0 | Illumina | WG-317-1003 |
| Illumina Infinium HumanMethylationEPIC v2.0 | Illumina | 20087709 |

### Participant recruitment, sample processing and ethics

Written informed consent was obtained from 349 study participants recruited as part of the Southern African Prostate Cancer Study (SAPCS, *n* = 308), with approval granted by the University of

Pretoria Faculty of Health Sciences Human Research Ethics Committee (HREC), with US Federalwide Assurance (FWA00002567 and IRB00002235 IORG0001762), in South Africa (HREC 43/2010) or the Ministry of Health and Social Services (MoHSS) in Namibia (17/3/3HEAF003). Additional Institutional Review Board (IRB) approval for SAPCS study recruitment was granted by the Human Research Protection Office (HRPO) of the US Army Medical Research and Development Command (E02371.2a TARGET Africa and E03333.1a and E05986.1a HEROIC PCaPH Africa1K in South Africa, and E03333.1b and E05986.1b HEROIC PCaPH Africa1K in Namibia). In Australia, St Vincent's Garvan Prostate Cancer Study participant recruitment ($n = 41$) was approved by St. Vincent's Hospital HREC (SVH/12/231). Recruitment occurred either at diagnosis (southern African) or surgery (Australian), with patients clinicopathologically classified using International Society of Urological Pathology (ISUP) grade grouping either with or without PCa (Table 1). All patients were treatment naïve at time of biopsy sampling.

Snap-frozen prostate cores (and patient-matched whole blood) were shipped on dry ice to the University of Sydney in Australia under a Republic of South Africa Department of Health Export Permit (National Health Act 2003; J1/2/4/2) and in accordance with institutional Material Transfer Agreement (MTA) and associated Collaborative Research Agreement (CRA). Irrespective of country of recruitment, all prostate tissue samples (245 SAPCS, 41 Australian) were processed and Qiagen-based DNA extracted at the University of Sydney (Hayes Lab), minimising for technical artefacts, with inclusion criteria based on availability of tissue samples with linked demographic and clinicopathological data. Tissue-associated Illumina Infinium HumanMethylationEPIC array data (Illumina, CA, USA) were generated at the Australian Genome Research Facility (AGRF, 2.8%) or the University of New South Wales Ramaciotti Centre for Genomics (Australia, 97.2%). Data generation and epigenomic/genomic interrogation were approved by the St. Vincent's Hospital HREC (SVH/15/227), the University of Pretoria Faculty of Health Sciences HREC (504/2022) and by the HRPO of the US Army Medical Research and Development Command (E02371 TARGET Africa; E03280.1a and E05984.1a HEROIC PCaPH Africa1K). Samples were excluded if they did not meet quality control criteria during DNA methylation data processing or due to low African tumour purity estimations, excluding 32.5% of samples (93/286, all SAPCS), leaving 193 (152 SAPCS, 41 Australian) for further analysis.

## Genomic resources and genetic ancestry fraction determination

As described previously (Jaratlerdsiri et al, 2022), blood-derived patient-matched whole genome germline small variant (SNVs and indels, MAF > 1%) data called for the 99 southern Africans (36 overlapping with 193 methylation-inclusive southern Africans) were made available via the SAPCS Data Access Committee (DAC) for generating a southern African EPIC-array SNP-filtering reference resource. To determine genetic ancestry for the 193 methylation-inclusive participants, 132,245 filtered and linkage disequilibrium (LD) pruned germline SNPs (MAF > 0.05, Hardy–Weinberg Equilibrium 1e-6 cutoff, LD pruned for window size of 50, step size of 10, $r^2$ threshold of 0.1), were made available to determine participant ancestry fractions (see Dataset EV34).

Additional reference population data were downloaded from the Human Genome Diversity Project (HGDP) and 1000 Genome Project (1KGP) subset of gnomAD v3.12 (Chen et al, 2023), including 20 CEU Europeans, 20 CHB Chinese, and non-matched patients from the Southern African Prostate Cancer Study (Jaratlerdsiri et al, 2022) (40 SAPCS). Using unsupervised ADMIXTURE v1.3.0 (Alexander et al, 2009) analysis ($K = 3$ replicated in 10 runs with average 0.524 cross-validation (CV) error), patient ancestries were assigned as "African" ($n = 119$, 2 Namibian, 117 South African) with >93% contribution (allowing for one Namibian outlier with 16% European fraction); "European" ($n = 48$, 18 Australian, 2 Namibian, 28 South African) if >90% contribution (allowing for a single outlier with an 11.1% Asian fraction) and "Asian" ($n = 23$, all Australian with >99% Asian fraction); with a single African-Asian admixed individual ($n = 1$ South African). A single African sample was excluded from estimations due to low coverage. Genetic ancestry for the 99 southern African reference participants (36 patient overlap) was determined using the same SNP set and reference data (Dataset EV35) and was replicated in 8/10 runs at $K = 3$, with a mean CV of 0.549. All patients were assigned "African" ancestry, with a majority ($n = 96$, 86 South African, 10 Namibian) with >87% African contribution, and three Namibian samples with between 14.9 and 22.7% European contribution. Namibians and South Africans were merged geographically for analyses as "southern African".

## Establishing southern African SNP-filtering resources

Our workflow for identifying probe overlap with African variants is shown in Fig. EV1. To produce the EPICv1 filtering resource, 99 individual African germline VCF files were lifted over from hg38 to hg19 genome build (Picard v2.21.9 LiftoverVcf (Van der Auwera and O'Connor, 2020)). Individual VCF files were then merged into a single consensus VCF file, filtered for only those variants that pass quality control filters (bcftools (Danecek et al, 2021) v1.9), and chromosome notation style converted to that of NCBI. The VariantAnnotation and GenomicRanges packages (Obenchain et al, 2014; Lawrence et al, 2013) were used to parse the single merged EPICv1 VCF file and extract all SNP and indel variants overlapping EPICv1 probes. As per Zhou et al's (Zhou et al, 2017) recommendations, we examined variants according to their probe position: (i) variants overlapping target CpG sites; (ii) variants overlapping single base extension (SBE) sites for Infinium type I probes; and (iii) variants overlapping the probe body within 5 bp from their targets. Results were filtered to include genetic variants with a maximum minor allele frequency >0.01 (Zhou et al, 2017) (see Datasets EV1–EV3). Barring lift over, the EPICv2 African filtering resource was generated using the same workflow and referencing the appropriate (hg38) Illumina manifest, excluding "chr0" and "chrM" probes (see Datasets EV4–EV6).

## Determining African-relevant CpG probe confounding

To determine the genomic distribution of African variant-confounding probe locations relative to specific features, EPICv1 probes were annotated using the Illumina manifest file "MethylationEPIC_v-1-0_B4.csv" mapped to hg19, and EPICv2 probes, using the "EPIC-8v2-0_A1.csv" manifest file mapped to hg38.

Where EPICv2 probe annotation was unknown, we used Gencode data Release 25 (GRCh38.p7), downloaded in December 2024 from https://ftp.ebi.ac.uk/pub/databases/gencode/Gencode_human/release_25/gencode.v25.annotation.gtf.gz, to extract gene body and intergenic regions using the GenomicRanges package. We defined intergenic regions as regions mapping outside known genes. Enhancer elements were downloaded in December 2024 from the FANTOM5 enhancer atlas: https://fantom.gsc.riken.jp/5/datafiles/latest/extra/Enhancers/ (hg19) and https://fantom.gsc.riken.jp/5/datafiles/reprocessed/hg38_latest/extra/enhancer/ (hg38), comprising 65,423 and 63,285 enhancers from 1829 human tissue samples, respectively (Lizio et al, 2015).

## Determining array version confounding

Matched African EPICv1 and EPICv2 replicate pairs were merged using RnBeads (v2.23.0) (Müller et al, 2019), identifying shared probes across platforms by genome coordinate position. After merging raw datasets, the correlation between 721,435 CpG sites common to both arrays was assessed for each replicate pair. For interrogation of array version as a confounder, raw EPICv1 ($n = 78$) and EPICv2 ($n = 115$) datasets were normalised using the "scaling.internal" method, background subtraction performed with the "sesame.noobsb" method, and datasets merged thereafter.

## Establishing discovery and validation methylation cohorts

Avoiding for array version confounding, tissue-derived Human-MethylationEPIC BeadChip v1.0 or v2.0 were analysed separately, allowing the larger resource to form the discovery arm and the smaller for validation. Consequently, the ancestry-associated discovery cohort consists of 70 EPICv1 samples and the validation cohort 57 EPICv2 samples. Conversely, the tumour-associated discovery cohort is made up of 93 EPICv2 samples and the validation cohort comprises 29 EPICv1 samples (Table 1). Only discovery cohort observations that were consistent with the validation cohort were presented.

## Array version-specific data processing, quality control and analyses

Using RnBeads and relevant annotation packages (hg19 for EPICv1 or hg38 for EPICv2, both v1.37.0), data were normalised using the "scaling.internal" method, followed by "sesame.noobsb" background subtraction. Samples were filtered for probes with detection $P$ values >0.05, probes with a bead count <3, default RnBeads SNP-overlapping probes, probes not in a CpG context and probes with missing values in >1% of samples. Cross-reactive probe filtering was performed manually as per Pidsley et al (Pidsley et al, 2016) for EPICv1, and Peters et al (Peters et al, 2024) for EPICv2. Probes located on sex chromosomes were retained. Samples were considered low quality if they displayed a missing probe fraction >0.2%.

Probes and samples not meeting quality control criteria were removed from subsequent analyses, with 717,828 probes and 109 samples in the EPICv1 dataset, and 794,018 probes and 173 samples in the EPICv2 dataset remaining for analyses. Filtering summaries are available in Dataset EV36. Normalised β-values were used for all subsequent statistical analyses.

### Cellular deconvolution of methylation data

To account for cell type heterogeneity, using the Houseman algorithm (Houseman et al, 2012) in RnBeads for reference-based deconvolution, we estimated cell type composition in bulk tumour samples, using DNA methylation profiles from sorted cell types as a reference (G. Gerhauser, personal communication). Cell type proportion estimates for 193 samples can be found in Dataset EV34. To limit compounded heterogeneity introduced by the genomically diverse African samples, only African tumour samples belonging to the highest tumour purity quartile were retained for further analyses ($n = 57$).

### Repetitive element, partially methylated domain (PMD) and chromatin state annotation of CpG sites

Annotation data for repetitive elements (Alu, LINE-1 and LTR) corresponding to the RepeatMasker database, available through AnnotationHub (Morgan and Shepherd, 2024), were downloaded using the REMP package (v.1.30.0) (Zheng et al, 2017). Annotations for common PMDs were retrieved from ref. (Zhou et al, 2018).

To overlay the regulatory context of CpG sites, we utilised a prostate tumour-derived chromatin state annotation of the human genome, as per ref. (Pomerantz et al, 2020). For simplification where relevant, "Active non-prostate lineage enhancer", "Active prostate lineage-specific enhancer", "Bivalent poised enhancer", "Primed non-prostate lineage-specific enhancer" and "Primed prostate lineage enhancer" were collectively classed as "enhancers"; and "Active non-prostate lineage promoter", "Active prostate lineage-specific promoter" and "Bivalent poised promoter", as "promoters".

### Identifying differentially methylated positions

For each cohort comparison, DMPs were identified by performing linear modelling with the limma package (v3.61.9) (Ritchie et al, 2015), while adjusting for covariates where possible. For ancestry-associated analyses, age and tumour purity were adjusted for in the discovery cohort, and age in the validation cohort. In the tumour-associated analyses, age, stromal and immune cell content were adjusted for, while age and immune cell content were adjusted for in the validation cohort. Multiple testing correction was performed with the Benjamini–Hochberg (BH) false discovery rate (FDR) method, with probes deemed significantly differentially methylated if they displayed an FDR cutoff of $P < 0.05$ and an absolute Δβ (i.e. absolute difference in mean methylation between two groups) ≥20% (ancestry-associated analysis) or ≥30% (tumour-associated analysis). Hierarchical clustering analysis was performed on significant DMPs using Euclidean distance and ward.D2 linkage.

### Identifying differentially methylated regions

DMRs were identified for each cohort comparison using matched regression models to those implemented in the respective DMP analyses. The package DMRcate (v3.1.9) (Peters et al, 2015) was used to identify DMRs using a bandwidth smoothing window of 1000 bp, a bandwidth scaling factor of 2 and a minimum of five consecutive CpGs to define DMRs. DMRs with a minimum smoothed FDR cutoff of $P < 0.05$ and an absolute Δβ of ≥10% (ancestry-associated analysis) or ≥20% (tumour-associated analysis) were considered significant.

### Enrichment analysis

Using the gsameth and gsaregion functions of the missMethyl package (v1.39.14) (Phipson et al, 2016), we identified Molecular Signatures

Database (MSigDB, v.7.1) (Subramanian et al, 2005) gene sets that were significantly enriched with methylation changes identified during DMP and DMR (both minimum $|\Delta\beta| \geq 10\%$) identification. MSigDB gene sets were downloaded in November 2024 from https://bioinf.wehi.edu.au/MSigDB/v7.1/. Using a subset of prostate (tumour)-derived TCGA-PRAD samples ($n = 309$), patient-matched methylation (Illumina HumanMethylation450) and gene expression (STAR - Counts) quantification data were downloaded in January 2025 using the TCGAbiolinks package's (v2.34.0) (Colaprico et al, 2016) GDCquery function and used to compute Pearson correlations between CpG beta and log2 expression values. Patient-matched TCGA-PRAD samples were included if treatment-naïve, primary prostate gland-derived adenocarcinomas ($n = 274$), or normal solid tissue ($n = 35$). To determine the regulatory impact of significant DMPs/DMRs, only related genes with a significant correlation (FDR $P < 0.05$, $|r| > 0.25$) or not covered on the 450k array, were included for enrichment analysis. For our tumour-associated enrichment analysis, we further subset TCGA-PRAD methylation data to only include individuals of "white" (European) race ($n = 249$, 217 primary solid tumours, 32 solid tissue normal), performed tumour *versus* normal differential methylation analysis using *limma*, while adjusting for age at diagnosis, and used resultant European-related DMPs (FDR $P < 0.05$, $|\Delta\beta| \geq 10\%$) to identify uniquely African tumour-associated differentially methylated sites limited to those covered on the 450k array, then used as input. The missMethyl package adjusts for the underlying distribution of probes on the array, with genes annotated to DMP and DMR probes compared to lists of genes in each curated MSigDB gene set to identify those that are statistically over-represented.

DMPs and DMRs were flagged for overlap with GeneHancer "Double Elite" (Fishilevich et al, 2017) regions, i.e., both GeneHancer elements and genes having associations derived from at least two information sources. Putative target genes for DMP- and DMR-overlapping enhancers were tested for gene set enrichment in MSigDB gene sets using the RITAN and RITANdata packages (Zimmermann et al, 2019). Using TCGA-PRAD expression data, detailed above, target genes were filtered for those considered expressed in the prostate, defined by a mean log2 expression value $\geq 2$ per gene. We defined the background as all GeneHancer "Double Elite" genes and considered terms with an FDR $P < 0.05$ as significant.

## Statistical analysis

All statistical analyses were performed using R software (v4.4.1), with sample size reliant on the availability of scarce African-derived prostate tissue and clinicopathologically matched non-African material. Data generation was randomised between the cohorts to avoid batch effects, while initially blinded to genetic ancestry, genetic testing allowed for ancestral confirmation and ancestrally defined downstream analyses. Cross-platform methylation correlation and tumour-associated $\Delta\beta$ concordance between discovery and validation cohorts were assessed using Pearson's correlation coefficient. Probes with an FDR $< 0.05$ were considered significantly differentially methylated. Relative to non-significant differentially methylated sites, significant DMP/DMR enrichment across various genomic contexts was calculated using Fisher's exact test. For repetitive element and CGI-related region boxplots, group means were compared using Wilcoxon rank-sum tests. Levene's test was used to assess differences in variance between groups. To assess the

significance of overlap between ancestry-associated discovery and validation DMPs and DMRs, permutation tests were performed using 10,000 randomly sampled probe or region sets (matched in number and array background) from the full EPICv2-tested probe set. Empirical $P$ values were computed as the proportion of random overlaps equal to or exceeding the observed.

## Data availability

R scripts for parsing and identifying EPIC probes that overlap southern African variants have been deposited at GitHub and can be accessed at https://github.com/j-craddock/African-variants-overlapping-EPIC-probes. Methylation data files generated in this study have been deposited in the in the Gene Expression Omnibus (GEO) under accession number GSE304293 (https://www.ncbi.nlm.nih.gov/geo/query/acc.cgi?acc=GSE304293). Ancestry informative variant data and SNP-filtering reference data have been uploaded to the European Variation Archive (EVA, www.ebi.ac.uk/eva) under accession code PRJEB94535 (https://url.au.m.mimecastprotect.com/s/hVI4CVARKgCg32GGqCzhxiEZMKw?domain=ebi.ac.uk). All other data supporting the key findings of this study are available within the article, Appendix and Expanded View documentation, with source data provided.

The source data of this paper are collected in the following database record: biostudies:S-SCDT-10_1038-S44320-025-00153-x.

## Peer review information

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

## Acknowledgements

We are forever grateful to the study participants for their contribution to this research, as well as both current and past clinical support staff who have contributed to the Southern African Prostate Cancer Study (SAPCS), Namibian-SAPCS and the Garvan Institute St Vincent's Prostate Cancer Biobank. We further acknowledge the SAPCS and Garvan-St Vincent's Biobank Managers Tumisang Mbeki (University of Pretoria, South Africa) and Anne-Maree Haynes (Garvan Institute of Medical Research, Australia), respectively, Dr Clare Stirzaker and Dr Elena Zotenko (Garvan Institute of Medical Research, Australia) for sharing their original code for the identification of EPIC probes overlapping genetic variants, Jue Jiang (Ancestry & Health genomics laboratory, University of Sydney, Australia) for providing germline variant calling, and Dr Matthew Freedman and Dr Xintao Qiu (Dana–Farber Cancer Institute, Harvard Medical School, USA) for sharing the prostate cancer ChromHMM annotation. Data was generated at the Australian Genome Research Facility (AGRF, 2.8%) or the University of New South Wales' Ramaciotti Centre for Genomics (97.2%) in Australia, with computational infrastructure support provided by the University of Sydney Informatics Hub. This study was supported by funding received from the U.S.A. Congressionally Directed Medical Research Programs (CDMRP) Department of Defense (DoD) Prostate Cancer Research Program (PCRP) Idea Development Award (PC200390, TARGET Africa to VMH) and a HEROIC Consortium Award (PC210168 and PC230673, HEROIC PCaPH Africa1K to VMH and MSRB), a U.S.A. National Institute of Health (NIH) National Cancer Institute (NCI) Award (1R01CA285772-01 to VMH); U.S.A. Prostate Cancer Foundation (PCF) Challenge Award (2023CHAL4150 to VMH) and National Health and Medical Research Council (NHMRC) Ideas Grants (APP2001098 to VMH and MSRB; APP2010551 to VMH). JC was further supported by the National Research Foundation of South Africa (PMDS22070633683) and VMH by the Petre Foundation via the University of Sydney Foundation, Australia. This study forms part of a PhD dissertation for JC.

## Author contributions

**Jenna Craddock**: Data curation; Software; Formal analysis; Investigation; Visualisation; Methodology; Writing—original draft; Writing—review and editing. **Pavlo Lutsik**: Software; Supervision; Methodology; Writing—review and editing. **Pamela X Y Soh**: Software; Methodology; Writing—review and editing. **Melanie Louw**: Resources; Data curation; Pathology review. **Md Mehedi Hasan**: Data curation; Sample processing. **Sean M Patrick**: Project administration. **Shingai B A Mutambirwa**: Resources; Project administration. **Phillip D Stricker**: Resources; Project administration. **Hagen E A Förtsch**: Resources; Project administration. **M S Riana Bornman**: Resources; Funding acquisition; Project administration. **Clarissa Gerhäuser**: Software; Supervision; Methodology; Writing—review and editing. **Vanessa M Hayes**: Conceptualisation; Data curation; Software; Supervision; Funding acquisition; Methodology; Project administration; Writing—review and editing.

Source data underlying figure panels in this paper may have individual authorship assigned. Where available, figure panel/source data authorship is listed in the following database record: biostudies:S-SCDT-10_1038-S44320-025-00153-x.

## Disclosure and competing interests statement

Hayes is a Member of Active Surveillance Movember Committee and received an honorarium from The Korean Urological Oncology Society for 2024 Annual Conference as a guest speaker.

## HEROIC PCaPH Africa1K

**Co-principal investigators** Gail S Prins[12] & Peter Mungai Ngugi[13]

**Developmental team leads** Weerachai Jaratlerdsiri[2,14], Winstar Mokua Ombuki[13], Daniel M Moreira[12] & Ikenna C Madueke[12]

**SAPCS Resource Working Group members** Maphuti Tebogo Lebelo[1,15], Tumisang M N Mbeki[1], Muvhulawa Obida[1,16], Martin Obida[16], Raymond A Campbell[17], Mulalo B Radzuma[7], Golda Stellmacher[9], Jessie Gamxamub[9] & Reginald M J Menoe[18]

**Genomics and Data Science Working Group members** Jue Jiang[2], Tingting Gong[2], Korawich Uthayopas[2], Kazzem Gheybi[2], Ruotian Huang[2], Kangping Zhou[2], Umuna Maendo[2,19], Massimo Loda[20], Giuseppe Nicolo' Fanelli[20], David C Wedge[10], Avraam Tapinos[10], Vivien Holmes[10], Robert G Bristow[10], Daniel S Brewer[11,21], Abraham Gihawi[11], Colin S Cooper[11,22], Rosalind A Eeles[22] & Zsofia Kote-Jarai[22]

**Other key Consortia members (alphabetical order)** Maria Argos[23], Irene Barnhoorn[24], Lynn Birch[12], Muriuki Elias Nyaga[25], Micah O Oyaro[13], Joyce Shirinde[1], Douglas I Walker[26], Edwin O O Walong[27], Githui Shila Wanjiku[13] & Margaret Quaid[23]

[12]Department of Urology, University of Illinois at Chicago, Chicago, IL, USA. [13]East Africa Kidney Institute, Department of Urology, University of Nairobi, Nairobi, Kenya. [14]Computational Genomics Group, Charles Perkins Centre, School of Medical Sciences, Faculty of Medicine and Health, University of Sydney, Camperdown, NSW 2050, Australia. [15]Department of Physiology, Faculty of Health Sciences, University of Pretoria, Pretoria, South Africa. [16]Tshilidzini Hospital, Shayandima, Thohoyandou, Limpopo, South Africa. [17]Department of Urology, University of Pretoria, Pretoria, South Africa. [18]Life Peglerae Hospital, Rustenberg, North West, South Africa. [19]Botswana International University of Science and Technology, Palapye, Botswana. [20]Department of Pathology and Laboratory Medicine, Weil Cornell Medicine, New York Presbyterian-Weill Cornell Campus, New York, NY, USA. [21]Earlham Institute, Norwich, UK. [22]The Institute of Cancer Research, London, UK. [23]Department of Environmental Health, Boston University School of Public Health, Boston, MA, USA. [24]University of Venda, Thohoyandou, Limpopo, South Africa. [25]Meru County Referral Hospital, Moi University, Meru County, Kenya. [26]Gangarosa Department of Environmental Health, Emory University, Atlanta, GA, USA. [27]Department of Pathology, University of Nairobi, Nairobi, Kenya.

# Expanded View Figures

**Figure EV1. Identifying EPICv1 and EPICv2 probes that overlap southern African polymorphic variants.**

Workflow for the identification of EPIC probes (v1.0 and v2.0) overlapping African SNV and indel variants, rendering filtering resources. Consensus probe filtering resources are based on germline variants (MAF > 0.01) from 99 southern African men. indel insertion and deletion, MAF minor allele frequency, SBE single base extension, SNV single-nucleotide variant, VCF variant call format.

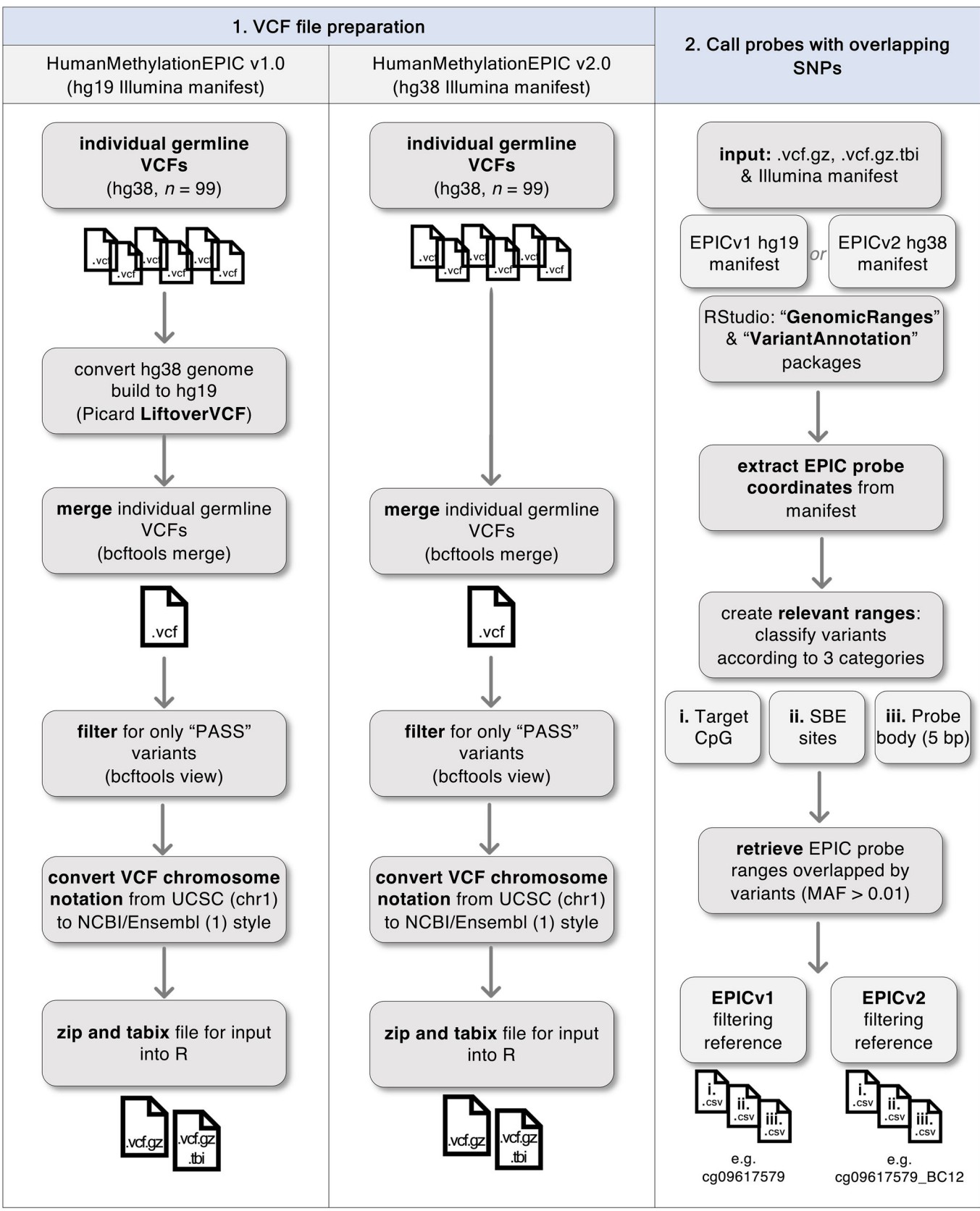

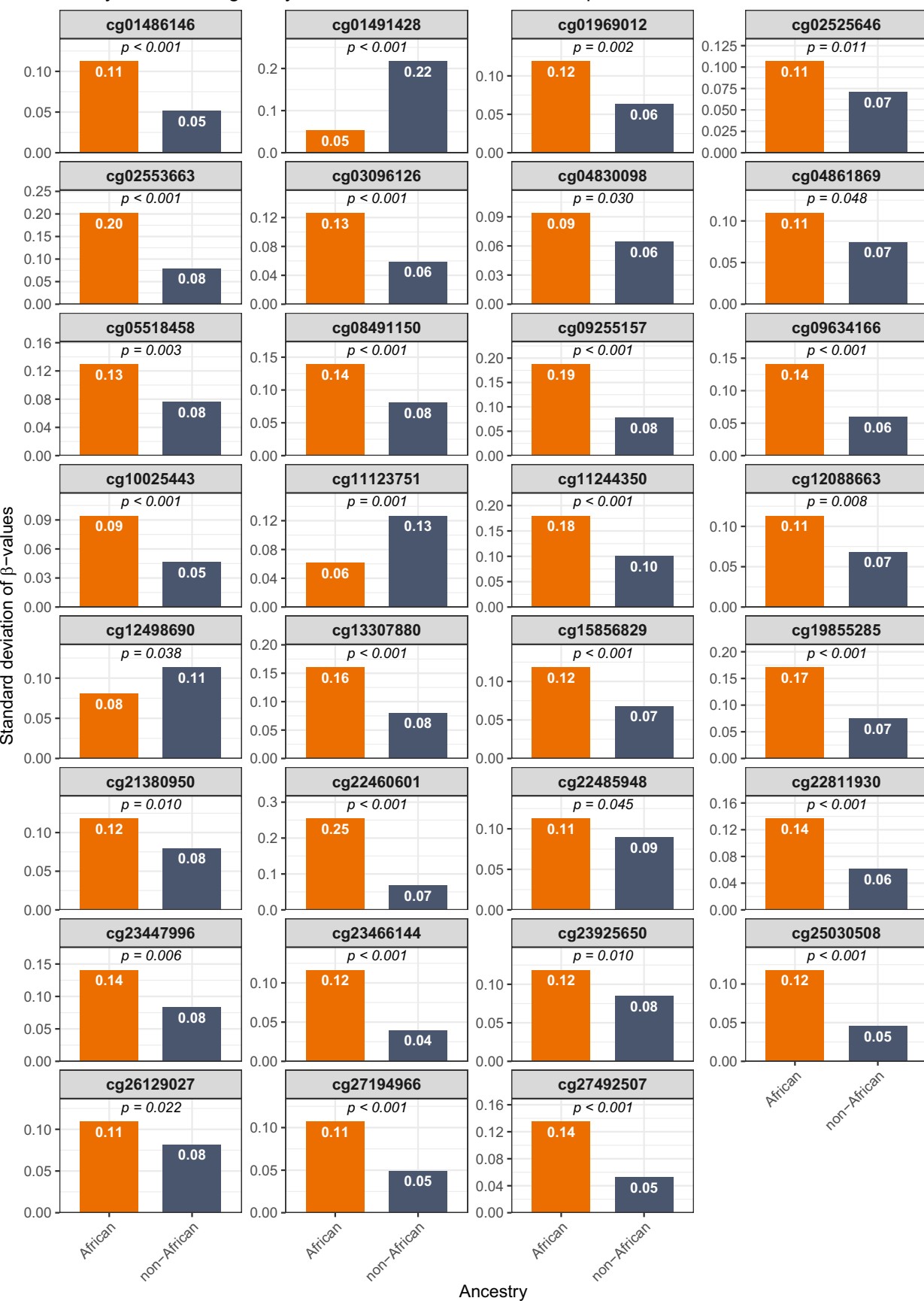

Methylation heterogeneity across African vs. non−African prostate tumours

**Figure EV2. African prostate tumours exhibit elevated methylation variability at ancestry-associated DMPs.**

Standard deviation of DNA methylation β-values is shown for each ancestry group at the 31 top DMPs exhibiting significant variance differences by ancestry (Levene's test, $P < 0.05$). Each panel presents within-group variability across African and non-African tumours. In 90.3% of these DMPs, African tumours displayed the highest intra-group variability, supporting the presence of elevated ancestry-associated epigenetic heterogeneity.

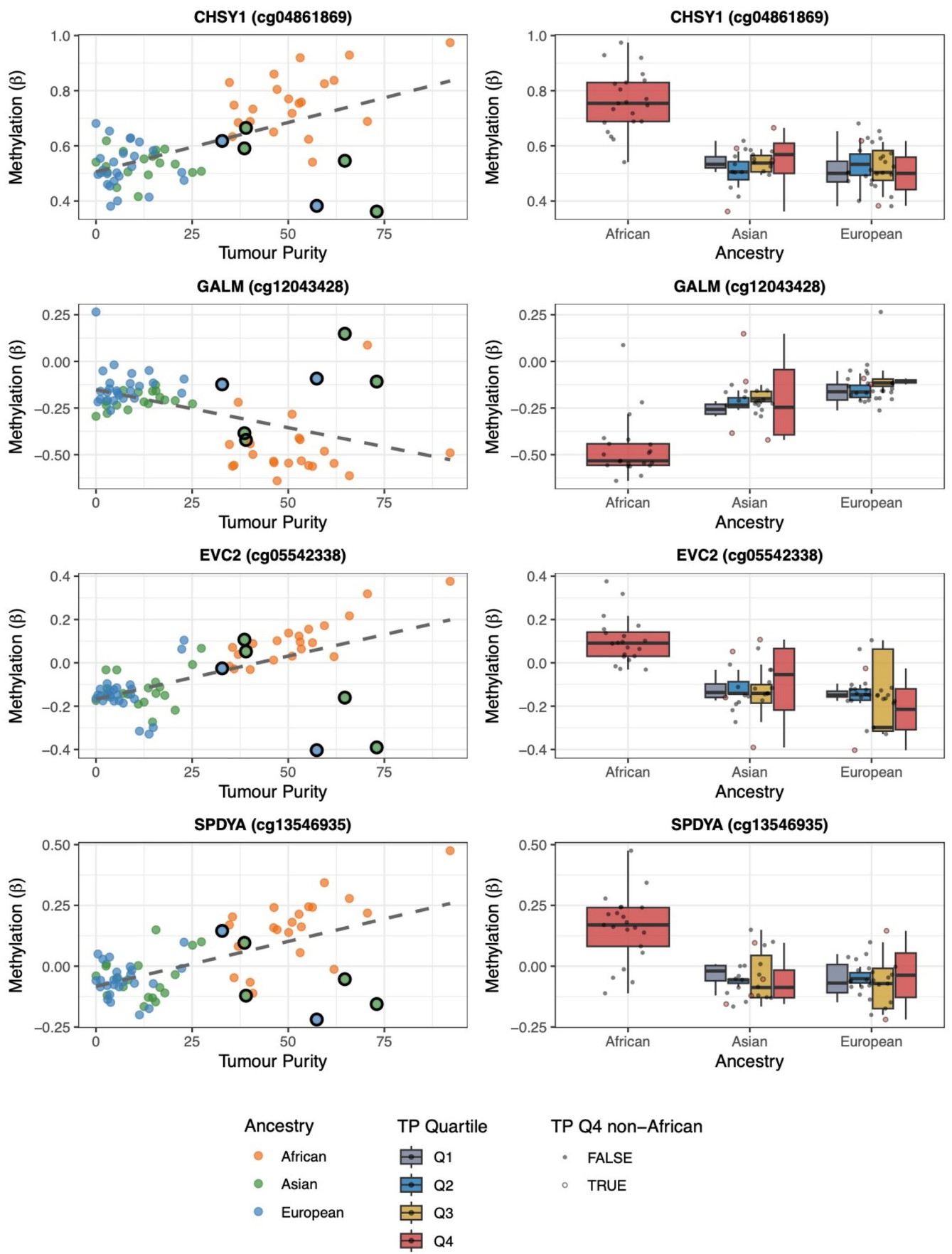

**Figure EV3. Ancestry-associated methylation patterns at four key DMP genes and their relationship to tumour purity.**

Scatterplots (left) and boxplots (right) depicting methylation levels at ancestry-associated CpG sites located in four genes of interest: *CHSY1*, *GALM*, *EVC2* and *SPDYA*. Scatterplots show methylation levels *versus* tumour purity across prostate tumours ($n = 70$), coloured by ancestry. Boxplots show ancestry-stratified methylation levels further grouped by tumour purity quartiles (Q1-Q4). Non-African Q4 tumour samples are outlined to highlight ancestry-related differences at high tumour purity.

