## [Peer Review File · Molecular Systems Biology]

Methylation reprogramming associated with aggressive prostate cancer and ancestral disparities

Jenna Craddock, Pavlo Lutsik, Pamela Soh, Melanie Louw, Md. Hasan, Sean Patrick, Shingai Mutambirwa, Phillip Stricker, Hagen Förtsch, MS Bornman, Clarissa Gerhauser, and Vanessa Hayes

Corresponding author(s): Vanessa Hayes (vanessa.hayes@sydney.edu.au)

Review Timeline:

Submission Date:	17th Apr 25
Editorial Decision:	5th Jun 25
Revision Received:	4th Aug 25
Editorial Decision:	2nd Sep 25
Revision Received:	9th Sep 25
Accepted:	17th Sep 25

Editor: Jingyi Hou

Transaction Report:

5th Jun 2025

Manuscript Number: MSB-2025-13061-T

Title: Methylation reprogramming associated with aggressive prostate cancer and ancestral disparities

Author: Vanessa Hayes

Jenna Craddock

Pavlo Lutsik

Pamela Soh

Melanie Louw

Md. Hasan

Sean Patrick

Shingai Mutambirwa

Phillip Stricker

Hagen Förtsch

MS Bornman

Clarissa Gerhauser

Dear Prof. Hayes,

Thank you for submitting your work to Molecular Systems Biology. We have now heard back from the three reviewers who agreed to evaluate your manuscript. As you will see from the reports below, the reviewers find the study and datasets generated interesting and relevant. However, they have also raised a series of important issues, which we would ask you to address in a major revision.

Without reiterating all the points raised, some of the more fundamental issues that need to be addressed include the following:

- Both Reviewers #1 and #3 raised concerns about potential confounding due to tumor purity. During our pre-decision cross-commenting process (in which editors invite reviewers to make additional comments on specific points), they offered constructive suggestions for addressing them. These additional comments are included below following their individual reports.
- Reviewer #2 had concerns regarding the presentation of the manuscript and requested a better integration of the results in the discussion. They also requested additional technical validation using orthogonal methods. These points should be carefully addressed.

All other issues raised by the reviewers need to be satisfactorily addressed as well. As you may already know, our editorial policy allows in principle a single round of major revision, so it is essential to provide responses to the reviewers' comments that are as complete as possible. Please feel free to contact me in case you would like to discuss in further detail any of the issues raised by the reviewers.

On a more editorial level, we would ask you to address the following issues:

- Please provide a .docx formatted version of the manuscript text (including legends for main figures, EV figures and tables). Please make sure that the changes are highlighted to be clearly visible.
- Please provide individual production quality figure files as .eps, .tif, .jpg (one file per figure).
- Please provide a .docx formatted letter INCLUDING the reviewers' reports and your detailed point-by-point responses to their comments. As part of the EMBO Press transparent editorial process, the point-by-point response is part of the Review Process File (RPF), which will be published alongside your paper.
- Please note that all corresponding authors are required to supply an ORCID ID for their name upon submission of a revised manuscript.
- We replaced Supplementary Information with Expanded View (EV) Figures and Tables that are collapsible/expandable online (see examples in <http://msb.embopress.org/content/11/6/812>). A maximum of 5 EV Figures can be typeset. EV Figures should be cited as 'Figure EV1, Figure EV2' etc... in the text and their respective legends should be included in the main text after the legends of regular figures.

Additional Tables/Datasets should be labeled and referred to as Table EV1, Dataset EV1, etc. Legends have to be provided in a separate tab in case of .xls files. Alternatively, the legend can be supplied as a separate text file (README) and zipped together with the Table/Dataset file.

For the figures and tables that you do NOT wish to display as Expanded View figures, they should be bundled together with their legends in a single PDF file called *Appendix*, which should start with a short Table of Content. Each legend should be below the corresponding Figure/Table in the Appendix. Appendix figures and tables should be referred to in the main text as: "Appendix Figure S1, Appendix Figure S2, Appendix Table S1" etc. See detailed instructions regarding expanded view here: <https://www.embopress.org/page/journal/17444292/authorguide#expandedview>.

-Before submitting your revision, primary datasets (and computer code, where appropriate) produced in this study need to be deposited in an appropriate public database (see [http://msb.embopress.org/authorguide - dataavailability](http://msb.embopress.org/authorguide-dataavailability) <https://www.embopress.org/page/journal/17444292/authorguide#dataavailability>). Please remember to provide a reviewer password if the datasets are not yet public. The accession numbers and database should be listed in a formal "Data Availability" section (placed after Materials & Method) that follows the model below (see also <https://www.embopress.org/page/journal/17444292/authorguide#dataavailability>). Please note that the Data Availability Section is restricted to new primary data that are part of this study.

Data availability

- RNA-Seq data: Gene Expression Omnibus GSE46843 (<https://www.ncbi.nlm.nih.gov/geo/query/acc.cgi?acc=GSE46843>)

- [data type]: [name of the resource] [accession number/identifier/doi] ([URL or identifiers.org/DATABASE:ACCESSION])

-At EMBO Press we ask authors to provide source data for the main figures. Our source data coordinator will contact you to discuss which figure panels we would need source data for and will also provide you with helpful tips on how to upload and organize the files.

- Our journal encourages inclusion of *data citations in the reference list* to directly cite datasets that were re-used and obtained from public databases. Data citations in the article text are distinct from normal bibliographical citations and should directly link to the database records from which the data can be accessed. In the main text, data citations are formatted as follows: "Data ref: Smith et al, 2001". In the Reference list, data citations must be labeled with "[DATASET]". A data reference must provide the database name, accession number/identifiers and a resolvable link to the landing page from which the data can be accessed at the end of the reference. Further instructions are available at .

- We updated our journal's competing interests policy in January 2022 and request authors to consider both actual and perceived competing interests. Please review the policy <https://www.embopress.org/competing-interests> and update your competing interests if necessary. Please use the heading "Disclosure statement and competing interests".

- All Materials and Methods need to be described in the main text using our 'Structured Methods' format. According to this format, the Methods section includes a Reagents and Tools Table (listing key reagents, experimental models, software and relevant equipment and including their sources and relevant identifiers) followed by a Methods and Protocols section describing the methods, ideally using a step-by-step protocol format. The aim is to facilitate adoption of the methodologies across labs.

Please download and fill our Reagents and Tools Table template (.docx), which you can find in our author guidelines: <https://www.embopress.org/page/journal/17444292/authorguide#structuredmethods>.

An example of a Method paper with Structured Methods can be found here: <https://www.embopress.org/doi/10.15252/msb.20178071>.

-Regarding data quantification:

Please ensure to specify the name of the statistical test used to generate error bars and P values, the number (n) of independent experiments (please specify technical or biological replicates) underlying each data point and the test used to calculate p-values in each figure legend. Discussion of statistical methodology can be reported in the materials and methods section, but figure legends should contain a basic description of n, P and the test applied.

Graphs must include a description of the bars and the error bars (s.d., s.e.m.).

- Please provide a "standfirst text" summarizing the study in one or two sentences (approximately 250 characters, including space), three to four "bullet points" highlighting the main findings and a "synopsis image" (550px width and 400-600 px height, PNG format) to highlight the paper on our homepage.

Here are a couple of examples:

<https://www.embopress.org/doi/10.15252/msb.20199356>

<https://www.embopress.org/doi/10.15252/msb.20209475>

<https://www.embopress.org/doi/10.15252/msb.209495>

When you resubmit your manuscript, please download our CHECKLIST (<https://www.embopress.org/pb-assets/embo-site/EMBO%20Press%20Author%20Checklist-1642513524327.xlsx>) and include the completed form in your submission.

Please note that the Author Checklist will be published alongside the paper as part of the transparent process (<https://www.embopress.org/page/journal/17444292/authorguide#transparentprocess>).

If you feel you can satisfactorily deal with these points and those listed by the referees, you may wish to submit a revised version of your manuscript. Please attach a covering letter giving details of the way in which you have handled each of the points raised by the referees. A revised manuscript will be once again subject to review and you probably understand that we can give you no guarantee at this stage that the eventual outcome will be favorable.

I look forward to receiving the revised manuscript soon.

Kind regards,
Jingyi

Jingyi Hou, PhD
Senior Editor
Molecular Systems Biology

*** PLEASE NOTE *** As part of the EMBO Press transparent editorial process initiative (see our Editorial at <https://dx.doi.org/10.1038/msb.2010.72>), Molecular Systems Biology publishes online a Review Process File with each accepted manuscripts. This file will be published in conjunction with your paper and will include the anonymous referee reports, your point-by-point response and all pertinent correspondence relating to the manuscript. If you do NOT want this File to be published, please inform the editorial office at contact@molsystbiol.org within 14 days upon receipt of the present letter.

Reviewer #1:

Summary

- Describe your understanding of the story

The incidence and severity of prostate cancer is highest in men of African ancestry and/or from Africa. DNA methylation changes are an established feature of prostate cancer. However, to date there have been few studies of DNA methylation in prostate tumour tissue from men of African ancestry, with studies of men living in Sub-Saharan Africa particularly lacking. This study seeks to redress this balance by performing a DNA methylation study on a cohort of prostate tumours from African men, as well as men of European and Asian ancestry for comparison. 30/48 of the tumours from men of European ancestry are from men from Africa, allowing the researchers to distinguish between ancestry versus geographical associations with methylation.

- What are the key conclusions: specific findings and concepts

The first outcome of the study was the development of a bioinformatic filtering resource to allow the processing of EPIC methylation microarray data for people of southern African ancestry.

In terms of biology, the authors find that prostate tumour methylation differences are primarily driven by ancestry rather than geographical differences between people. They also identify African specific prostate cancer methylation changes in gene pathways related to metastatic growth and disease aggressiveness.

- What were the methodology and model system used in this study

For the primary analysis of prostate cancer DNA methylation, prostate tissue was collected from n= 245 participants from the Southern African Prostate Cancer Study (SAPCS) and n=41 participants from the St Vincent's Garvan Prostate Cancer Study. DNA was extracted from snap-frozen prostate cores and run on the Illumina EPIC methylation microarray.

After removing African tumour samples of low tumour purity, the final prostate analysis was performed on n=193 samples comprising the following ancestries.

SAPCS: n= 57 PCa African, n= 65 non-PCa African, n=30 PCa European (total = 152)
Australian: n=18 PCa European, n=23 PCa Asian (total = n=41)

General remarks

- Are you convinced of the key conclusions?

The authors should be commended for considering at the outset the impact of African-specific SNVs on probe-cross hybridisation. The approach they have taken is very thorough. The resource they have built is not only important for the current study, but will also improve the work of others using the EPIC family of methylation microarrays for research including southern African populations. Overall, this will lead to better quality results by reducing technical artefacts and thus more equitable research.

The approach to splitting analyses into discovery and validation based on the array platform version (EPICv1 vs EPICv2) is a sensible solution to the unavoidable batch effect problem. The inclusion of the 'presumably PCa' samples as tumour samples also seems reasonable.

However, the removal of lower purity samples in only the African samples is concerning (as described in line 151-156), as it means any comparisons between ancestry groups is confounded by differences in tumour purity. See major points below for details.

In general the methods used for downstream analysis characterising the DMPs and DMR are very thorough, and background content of the arrays has been well-controlled for. It is a clever idea to use an expression dataset to filter for functional methylation changes for pathway analysis.

Place the work in its context.

- What is the nature of the advance (conceptual, technical, clinical)?

The main advance of the paper is the technical advance of creating a workflow and resource for a technically sound analysis of methylation from people with southern African genetic ancestry.

The other advance is in the curation of prostate methylation data from a cohort of men with prostate cancer of southern African ancestry and public release of this data as a resource to other prostate cancer epigenetics researchers.

- How significant is the advance compared to previous knowledge?

The creation of a southern African specific workflow is an advance for the epigenetics research community. The concept of accounting for population specific SNPs is well-established, so the novelty here lies in using SNP data from 99 people from the SAPCS study to tailor this quality control step to southern Africans.

Whilst there have been many studies of DNA methylation in prostate cancer, the current study is novel in generating prostate tumour methylation from men of southern African ancestry, a population that is poorly represented in current cancer epigenetic research.

- What audience will be interested in this study?

This work would be of interest to prostate cancer researchers interested in the molecular mechanisms that may underlie why prostate cancer is more aggressive in men of African ancestry. Epigeneticists may also be interested in ancestry-specific methylation signatures in the normal tissue.

Major points

- Specific criticisms related to key conclusions

- The main criticism is that the methylation data from the African samples were selected to have high tumour purity, whilst the same selection criteria was not applied to other ancestry groups, creating a significant confound for the data analysis. As aggressive tumours tend to have a higher tumour load this is likely why the authors "found African prostate tumours to exhibit distinct methylation patterns akin to aggressive disease" (line 580-581).

Indeed, in Table 1 the estimated proportion of luminal cells in African PCa samples is 49.6% compared to 13.1% in European samples and 19.2% in Asian samples. In Supplementary Table 1 cell composition is much more significantly associated with PC1 than ancestry. It is therefore impossible to distinguish whether ancestry methylation differences are in fact differences in cell composition between groups. A difference in immune cell proportions may be why downstream analyses shows an association between hypomethylated 'ancestry-DMPs' and immune-related gene sets.

In line 200-201 the authors claim that 'African tumours showed great between-sample heterogeneity not adequately explained by tumour content'. However, tumour content may largely explain the heterogeneity - the data in Supplementary Figure 5 shows that the x-axis (PC1) is associated with T-luminal content, whilst the variation along the y-axis (PC2) is likely associated with stromal and immune cell content according to Supplementary Table 1.

- Relating to line 209-210 - as clustering of Figure 2b was done specifically on the 861 ancestry-DMPs it is unsurprising that the samples in this heatmap do not cluster on geography. To identify geography DMPs a separate analysis using geography as the variable of interest would be required. It is also not clear how Figure 2b shows the stated 'extensive African-specific tumour heterogeneity' - The methylomes for each probe (row) in this plot look very uniform within the samples of men with African ancestry (as would be expected of a probe identified as an ancestry-DMPs).

- There is little overlap between African DMPs and TCGA DMPs - could this again be due to purity differences between the 2 cohorts. Do African DMPs trend in the same direction in the TCGA data? This could be assessed by a correlation of delta beta (tumour-normal methylation differences) between each dataset.

-There is a surprising lack of overlap between the discovery and validation analyses in Figure 2g. What could explain this? Do the 861 DMPs trend in the same direction but not reach statistical cut-offs? Given the smaller sample size in the validation cohort, would a targeted analysis of just the 861 DMPs (reducing the multiple testing burden) provide a better assessment of the methylation change at these loci in the validation cohort.

-Specify experiments or analyses required to demonstrate the conclusions

-Motivate your critique with relevant citations and argumentation

Further work is needed to demonstrate that the ancestry methylation differences are not just differences in tumour purity. The best way to address this would be to include the methylation data from the samples that were excluded from the SAPCS cohort due to low tumour purity in all analyses.

Another approach would be to look specifically at the top probes in table 2 and perform correlation analysis and visualisations to compare their methylation levels with tumour purity across all tumour samples in the current study. The few European and Asian samples with high tumour purity could be highlighted in scatter plots or box plots to show that the difference is truly between ancestry groups and not just driven by tumour purity. Likewise, the boxplots in supplementary figure 6 could be repeated but with the data grouped by tumour purity.

I appreciate that in some cases cell composition measures have been used in the regression models to identify DMPs, however is their use statistically valid given that they are highly correlated with the ancestry variable? Please discuss.

To identify whether ancestry DMPs are tumour specific the normal prostate tissue methylation data should be added to the heatmap in figure 2b.

A separate analysis of the geographic difference within men of African ancestry would be needed to show that ancestry differences are more substantial than geographical differences. This would also help to address the question of whether environmental effects would be playing a role in prostate cancer, as suggested in the introduction.

Minor points

-Easily addressable points

- It would be helpful to point out that PC2 in the y-axis in Fig 1f only represents 5% of the variance in the dataset - so although there is a clear batch effect due to the different versions of the microarrays used, it is a relatively small source of variance compared to ancestry in PC1 - i.e. the main source of variance in the dataset is driven by biological rather than technical factors

- Why were different Beta value cut-off thresholds used for difference analysis - i.e. 20% for ancestry analysis; 30% for tumour-normal analysis; 10% for TCGA tumour-normal analysis. Was it for practical reasons such as having a manageable number of probes for downstream analysis in each case? Please also explain why different combinations of covariates were used for different analyses (age, tumour purity, stromal and immune cell content).

- In cases where groups are unbalanced it would be preferable to use Wilcoxon tests rather than t-tests, e.g. for the analysis accompanying supp figure 6.

-Presentation and style

NA - figures are beautifully presented

-Trivial mistakes

Figure 4d - it looks like the key label for cancer status is the wrong way around. i.e. the dark purple should be PCa

Reviewer #2:

Major Strengths

1. Really nice to see more research into non-European cohorts, definitely a strength of this paper, the lack of research into more

ethnically diverse cohorts has been a long-standing weakness in cancer research to date.

2. Rigorous Technical Methodology. The authors demonstrate great attention to technical rigor in their analysis. Clear sense of the efforts of the research team to ensure all boxes were ticked and controlled for. Cross platform analysis, controlling for SNPs, comments on Type I vs Type II content - all great to see.

Major Weaknesses

1. Predominantly Descriptive Analysis with Limited Biological Contextualization

While the technical analysis is comprehensive and methodologically sound, the manuscript suffers from what seems to be an overemphasis on observational descriptions, with deeper context or meaning frequently lacking, regarding biological interpretation or mechanistic insight, and/or the relevance pertaining to why some results are being commented on.

For example (pg 19, lines 340-344).

"tumour associated DMPs were primarily located in non-CGI regions (72.21%), followed by CGIs and shores (21.76%), while showing predominant enrichment in heterochromatin (34.12%, $p = 4.33 \times 10^{-13}$), followed by promoters (30.88%), including active non-prostate (12.16%, $p < 2.2 \times 10^{-16}$) and prostate lineage-specific (2.21%, $p < 2.2 \times 10^{-16}$), and bivalent poised (16.51%, $p < 2.2 \times 10^{-16}$) promoters, and enhancers (24.26%, $p = 2.51 \times 10^{-12}$)".

While the observations above are completely valid, my question is - what is the relevance of these findings? Why are the authors mentioning bivalent promoters specifically here (just as an example)? If it's being mentioned in this very granular list of genomic loci, it suggests this result should somehow relate to key differences underscoring ancestry differences in PCa. But bivalent promoters (as an example) aren't mentioned anywhere in the discussion, nor expanded upon elsewhere in the analysis.

Which begs the question - what is the key message, and why list so much superfluous information?

This is a frequent weakness in bioinformatically based papers, and my recommendation is always the same: if the information mentioned in results is just listing the top X number of statistically high hits from an intersection, but that information doesn't relate to the key conclusions of that experiment, it doesn't enhance understanding for the reader about the biological question under investigation.

This is just one example, so the authors shouldn't focus necessarily on rebutting this specific section I've highlighted. But throughout the manuscript results are presented in very minute detail, and in each case I would challenge to ask - *why* is this relevant to my understanding?. *How* do the results deepen our understanding of the underlying biology of PCa, ancestry, and DNA methylation.

Similarly, the discussion tends towards some general comments about broad pathways, but frequently fails to delve into the specifics within their own dataset.

For example, from DISCUSSION pg 26 lines 494-500.:

"Analogous to our ancestry associations, African-derived tumours over non-tumours were primarily hypermethylated, with DMPs biased to non-CGI regions and displaying substantial enhancer distribution. Enhancers regulate gene expression distally via interactions with TFs and transcriptional machinery, and DNA looping to achieve proximity with gene promoters. Cancer significantly alters enhancer methylation, affecting (cancer-associated) target gene expression²⁵. We again observed target gene enrichment for "CHANDRAN_METASTASIS_DN", indicating both direct gene dysregulation through aberrant methylation and indirect effects via epigenetically altered interacting enhancers."

The section starts by making a broad comment about their data ("non-tumours were primarily hypermethylated") but the immediately moves into a commentary about enhancers that is so broad, it could relate to any dataset.

As the paper is framed in the context of ancestry, (and is the main innovation of the paper) a deeper integration of their results in the discussion is required, with references back to the main figure in the results so a reader can identify the data being referred to and its relevance.

2. Absence of Technical Validation Using Orthogonal Methods

There should be some technical validation of DMRs using an orthogonal method, for example by targeted bisulfite

resequencing. Given the differences observed between array probe types, the smaller cohort size, and other sources of variation and error (i.e., lot variations in the arrays), validation of their results by an orthologous method is required.

As a wet lab method, DNA methylation analysis using targeted resequencing is relatively easy and scales well to working with 2-3 plates of samples, and would be cost effective with a small panel of 50-100 regions (e.g., a single Nextseq 2000 Mid kit run using 1x150bp reads). While some validation with other publicly available data was done, technical validation on the same samples is needed to confirm the key findings regarding differences in DNA methylation. There's a tendency of the field to take beta values from the arrays as gospel, but the technical precision of the beadchip arrays isn't great, and I've seen firsthand how methylation values from arrays don't hold up when validated by sequencing - a problem of some significance when sample sizes get smaller (as they start to in this manuscript)

Minor Comments

1. Would suggest also adding in a comment that the variation could be attributable to lot differences in array version (pg5 lines 129-135). Historically for our EPIC studies we try to request Illumina ship arrays from the sample lot to control for this.

2. Conservative Approach to Sample Reclassification Recommended

I'm not enthusiastic about the decision of the authors to reclassify 7 non-tumour to "presumably PCa" based on methylation data alone, since it's something of a logical leap to discount the pathology histopathology classification based entirely base just on the methylation data.

It would have been better to exclude them fully from the analysis, or at least analyze them separately. My reading of the paper is that they were lumped in with the rest of the PCa samples, and it's difficult to infer what effect these 7 samples could have, particularly when numbers start to dwindle on the subgroup analyses.

I won't request re-analysis, but the authors needs to revise Table1 and show how many samples in a subgroup contain these 7 outliers (and also note this elsewhere in other figures, when this 7 are part of the analysis set).

For example, the Ancestry-associated cohort has

Discovery = 21

Validation = 35.

The table needs to be revised to show (for example, just making up numbers):

Discovery = 21 (6 = presumably PCa)

Validation = 35 (1 = presumably PCa)

3.

Their figure6 is a nice snapshot of the key conclusions, but the figure title needs to be changed to make it clear this is the author's interpretation and proposed model, as there's no functional data in their study that definitively establishes this is mechanistically what is happening. Simply revising to say "Proposed Model....etcetc" would be sufficient.

Recommendation

Work that is both interesting to the field and relevant, but the current manuscript has some notable gaps as outlined above.

Given my recommendation to generate supporting validation data, either major revision, or opportunity to resubmit, would both be appropriate.

Reviewer #3:

The authors are addressing a key gap in cancer omics research: the lack of data covering the great heterogeneity of African populations. With a rather large sample size (nearly 200), this dataset will be a great resource for future prostate cancer (PCa) studies, and will probably be cited heavily.

The analysis is as rigorous as the current state of the art allows. The authors spent significant efforts identifying confounding CpGs in the methylation chip, and systemically assessed the confounding effects of 2 versions of the chip. They also included

tumor purity information in the majority of figures, making confounding factors very obvious.

That being said, the conclusions are still confounded by tumor purity. However, this is not a limitation of this manuscript per se, but a limitation of the field as it is. When comparing tumor to non-tumor, for example, non-tumor samples will inevitably have higher stromal fractions. I appreciate that the authors were very transparent about this. Although a few deconvolution methods have been developed that aim to computationally split the data into tumor-only and stromal-only, none of them seem to work very well from my experience. In short, this limitation should not diminish the quality and value of this dataset itself. It just means that this dataset will be reanalyzed in the future once better methodologies have been developed.

A related discussion is whether it is appropriate to filter African PCa samples to only use those with high tumor purity. This selection procedure induces bias since African samples will have higher purity than European/Asian samples. However, without selection, the comparison would be "impure samples vs impure samples", which is not meaningful. The authors acknowledged the bias induced by the selection, which is rigorous. Importantly, they were able to observe that the great heterogeneity of African samples can not be explained by purity.

Major revisions:

1. Visually, Fig. 2b doesn't correlate well with Fig. 2c. Fig. 2c shows that the absolute methylation differences of the DMPs are at least 0.2, but this is not visible for the majority of rows shown in Fig. 2b. This could be due to the color scale. It is important to adjust this since the heatmap helps the reader visualize the consistency/heterogeneity within each ancestry. One possible method is to include a row-scaled z-score-based heatmap side-by-side.

2. In Fig. 2g, the validation cohort showed many more DMPs and DMRs (16 times more DMPs, for example). The authors need to explain how this key inconsistency occurred. The authors also need to calculate if such overlap is significant over a random background, given the large number of DMP/DMRs in the validation cohort. In addition, the Venn plot is misleading since the area of overlap does not reflect the percentage of overlap.

3. For the key genes reported in the manuscript (GALM, EVC2, CHSY1, and SPDYA; PYCARD and SLC12A9), additional validation is needed to ascertain how much they are confounded by tumor purity. The authors could take advantage of public datasets (if any) that profiled the methylation levels of these loci in purified luminal, stromal, and immune cells.

Minor issues:

1. Fig. 1a-d needs an explicit title. It feels like the distribution of all probes, not the probes that are confounded.

2. Line 210: "extensive African-specific tumour heterogeneity (Fig. 2b)". Not sure what this is referring to. I did not observe any heterogeneity in Fig. 2b

Pre-decision cross-commenting

Reviewer #3

As the more "optimistic" reviewer #3, I do understand the points made by reviewer #1. Two approaches to address this issue were brought up: 1) reanalyze the data with all samples (including the impure ones); 2) for the top discoveries listed in Table 2, rule out the possibility that they are confounded by purity.

I strongly favor the second approach. Although approach 1) will lead to a more apples-to-apples comparison, interpreting the results might be challenging, since it will be impure African samples vs impure European/Asian samples. By contrast, approach 2) is critical, since Table 2 contains the key findings of this paper that could be cited in the field widely, necessitating extra efforts to make sure they are not strongly confounded. I also raised a similar point in Major Point 3 for the key genes reported in the manuscript (GALM, EVC2, CHSY1, and SPDYA; PYCARD and SLC12A9).

Reviewer #1

I agree with reviewer 3's new comment that additional analysis of the top discoveries listed in Table 2 is a good compromise, and thus I still recommend the straightforward analyses recommended in my original review.

i.e. 'look specifically at the top probes in table 2 and perform correlation analysis and visualisations to compare their methylation levels with tumour purity across all tumour samples in the current study. The few European and Asian samples with high tumour purity could be highlighted in scatter plots or box plots to show that the difference is truly between ancestry groups and not just driven by tumour purity. Likewise, the boxplots in supplementary figure 6 could be repeated but with the data grouped by tumour purity.

Manuscript Number: MSB-2025-13061-T

Title: Methylation reprogramming associated with aggressive prostate cancer and ancestral disparities

Response to Reviewers Comments

Reviewer/editor comments: Black

Author responses: Blue

Main text updates: Red

Reviewer #1

Summary

- **Describe your understanding of the story**

The incidence and severity of prostate cancer is highest in men of African ancestry and/or from Africa. DNA methylation changes are an established feature of prostate cancer. However, to date there have been few studies of DNA methylation in prostate tumour tissue from men of African ancestry, with studies of men living in Sub-Saharan Africa particularly lacking. This study seeks to redress this balance by performing a DNA methylation study on a cohort of prostate tumours from African men, as well as men of European and Asian ancestry for comparison. 30/48 of the tumours from men of European ancestry are from men from Africa, allowing the researchers to distinguish between ancestry versus geographical associations with methylation.

- **What are the key conclusions: specific findings and concepts**

The first outcome of the study was the development of a bioinformatic filtering resource to allow the processing of EPIC methylation microarray data for people of southern African ancestry.

In terms of biology, the authors find that prostate tumour methylation differences are primarily driven by ancestry rather than geographical differences between people. They also identify African specific prostate cancer methylation changes in gene pathways related to metastatic growth and disease aggressiveness.

- **What were the methodology and model system used in this study**

For the primary analysis of prostate cancer DNA methylation, prostate tissue was collected from n= 245 participants from the Southern African Prostate Cancer Study (SAPCS) and n=41 participants from the St Vincent's Garvan Prostate Cancer Study. DNA was extracted from snap-frozen prostate cores and run on the Illumina EPIC methylation microarray.

After removing African tumour samples of low tumour purity, the final prostate analysis was performed on n=193 samples comprising the following ancestries.

SAPCS: n= 57 PCa African, n= 65 non-PCa African, n=30 PCa European (total = 152)

Australian: n=18 PCa European, n=23 PCa Asian (total = n=41)

General remarks

- **Are you convinced of the key conclusions?**

The authors should be commended for considering at the outset the impact of African-specific SNVs on probe-cross hybridisation. The approach they have taken is very thorough. The resource they have built is not only important for the current study, but will also improve the work of others using the EPIC family of methylation microarrays for research including southern African populations. Overall, this will lead to better quality results by reducing technical artefacts and thus more equitable research.

Response: We appreciate the acknowledgement of the study design and collaborative effort of the entire SAPCS team to establish this critical resource.

The approach to splitting analyses into discovery and validation based on the array platform version (EPICv1 vs EPICv2) is a sensible solution to the unavoidable batch effect problem. The inclusion of the 'presumably PCA' samples as tumour samples also seems reasonable.

Response: Appreciated.

However, the removal of lower purity samples in only the African samples is concerning (as described in line 151-156), as it means any comparisons between ancestry groups is confounded by differences in tumour purity. See major points below for details.

Response: Please see detailed response under Major Points.

In general the methods used for downstream analysis characterising the DMPs and DMR are very thorough, and background content of the arrays has been well-controlled for. It is a clever idea to use an expression dataset to filter for functional methylation changes for pathway analysis.

Response: Appreciated.

Place the work in its context.

- **What is the nature of the advance (conceptual, technical, clinical)?**

The main advance of the paper is the technical advance of creating a workflow and resource for a technically sound analysis of methylation from people with southern African genetic ancestry.

The other advance is in the curation of prostate methylation data from a cohort of men with prostate cancer of southern African ancestry and public release of this data as a resource to other prostate cancer epigenetics researchers.

- **How significant is the advance compared to previous knowledge?**

The creation of a southern African specific workflow is an advance for the epigenetics research community. The concept of accounting for population specific SNPs is well-established, so the novelty here lies in using SNP data from 99 people from the SAPCS study to tailor this quality control step to southern Africans.

Whilst there have been many studies of DNA methylation in prostate cancer, the current study is novel in generating prostate tumour methylation from men of southern African ancestry, a population that is poorly represented in current cancer epigenetic research.

- **What audience will be interested in this study?**

This work would be of interest to prostate cancer researchers interested in the molecular

mechanisms that may underlie why prostate cancer is more aggressive in men of African ancestry. Epigeneticists may also be interested in ancestry-specific methylation signatures in the normal tissue.

Response: We greatly appreciate the reviewers' comments to the study significance and relevance.

Major points

- **Specific criticisms related to key conclusions**

1. The main criticism is that the methylation data from the African samples were selected to have high tumour purity, whilst the same selection criteria was not applied to other ancestry groups, creating a significant confound for the data analysis. As aggressive tumours tend to have a higher tumour load this is likely why the authors "found African prostate tumours to exhibit distinct methylation patterns akin to aggressive disease" (line 580-581).

Indeed, in Table 1 the estimated proportion of luminal cells in African PCa samples is 49.6% compared to 13.1% in European samples and 19.2% in Asian samples. In Supplementary Table 1 cell composition is much more significantly associated with PC1 than ancestry. It is therefore impossible to distinguish whether ancestry methylation differences are in fact differences in cell composition between groups.

Response: Confounding of data due to tumour purity and cell content selection

We thank the reviewer for this observation. We agree that cell composition, particularly T-luminal content, is a major contributor to DNA methylation variability and strongly associated with PC1 in the full dataset (Supplementary Table 1, now renamed Table EV1). However, ancestry-associated analyses were conducted on a filtered discovery subset ($n = 70$), for which Table EV2 is the relevant reference. In this cohort, African samples were restricted to those in the highest tumour purity quartile ("TP Q4") to reduce compositional confounding. This filtering lessened the contribution of T-luminal content ($p = 4.7 \times 10^{-33}$ to 1.3×10^{-12}) and strengthened the ancestry association ($p = 2.1 \times 10^{-4}$ to 1.9×10^{-9}), alongside an increase in PC1 variance explained from 59% to 78%. These changes indicate that filtering reduced non-biological variability and enhanced detection of ancestry-associated methylation differences.

While tumour purity remains a significant covariate, it is not fully collinear with ancestry, as evidenced by the presence of high-purity samples across all ancestry groups (Supplementary Fig. 5, now renamed Appendix Fig S4). This enabled statistical adjustment for tumour purity in differential methylation analyses. The persistence of significant ancestry-associated DMPs and DMRs after adjustment reinforces the conclusion that ancestry exerts an independent influence on the tumour methylome. Applying the same "TP Q4" filter to non-Africans would have reduced the group to just six individuals, undermining statistical power. Instead, we opted for a hybrid approach: stringent filtering of African samples and covariate adjustment across all samples. This allowed for comparisons across mid-to-high purity tumours while controlling for key confounders. We also note that PCA reflects unadjusted methylation data, which visually accentuates tumour purity effects. Again, our downstream models accounted for this confound, enabling identification of ancestry-associated epigenetic features.

Lastly, lower-purity African tumours were excluded because their inclusion introduced excessive variability, likely reflecting both increased cellular admixture and underlying inter-individual methylation diversity, which masked ancestry-associated signal in regression models. By focusing on high-purity samples and adjusting for purity, we improved resolution of ancestry-specific differences while minimising noise.

A difference in immune cell proportions may be why downstream analyses shows an association between hypomethylated 'ancestry-DMPs' and immune-related gene sets.

Response: We acknowledge that PC2, which accounts for 4.1% of total variance, is associated with stromal and immune cell content. However, PC1, the dominant axis of variation (78.8% in the filtered ancestry cohort), is primarily driven by ancestry and tumour purity. As such, the major ancestry-associated methylation signal is unlikely to be largely attributable to immune cell composition. While we considered adjusting for stromal and immune content, these variables were excluded from differential methylation models due to their limited contribution to overall variance, which would have increased the risk of overfitting. We acknowledge that some residual immune-associated signal may persist, but the reproducible enrichment of immune-related gene sets across both discovery and validation cohorts (despite differences in sample composition, array version and statistical models) suggests this may represent a consistent and biologically relevant feature of ancestry-associated prostate tumour methylation. We've updated the discussion in the revised manuscript to state that this is a limitation and to advise careful interpretation.

In line 200-201 the authors claim that 'African tumours showed great between-sample heterogeneity not adequately explained by tumour content'. However, tumour content may largely explain the heterogeneity - the data in Supplementary Figure 5 shows that the x-axis (PC1) is associated with T-luminal content, whilst the variation along the y-axis (PC2) is likely associated with stromal and immune cell content according to Supplementary Table 1.

Response: We appreciate the reviewer's point and agree that tumour purity contributes significantly to variance in DNA methylation, particularly in PCA. However, African tumours remained slightly more dispersed across PC1, even after restricting to "TP Q4" samples, than non-African tumours of comparable purity. This pattern, visible in both panels of Appendix Fig S4, indicates that cell composition alone does not fully explain the observed between-sample variability. Our approach sought to minimise compositional confounding while retaining biologically meaningful ancestry-associated signal. We acknowledge the inherent challenges in disentangling biological variability from tumour microenvironment effects, particularly in ancestry-structured methylation studies, and have revised the manuscript to reflect these limitations and interpretive boundaries.

However, to formally evaluate methylation heterogeneity across ancestry groups, we performed Levene's test on the top 100 differentially methylated positions (DMPs), testing for differences in β -value variance across African and non-African prostate tumours. This analysis revealed that 31 of 100 DMPs showed significantly different variances by ancestry ($p < 0.05$), with 26 remaining significant following FDR correction. To assess the direction of this heterogeneity, we computed the standard deviation of methylation values within each ancestry group for the 31 significant probes. Notably, in 28 of the 31 DMPs (90.3%), the African group displayed the highest intra-group variability (Figure EV2). This supports the

conclusion that African prostate tumours exhibit elevated epigenetic heterogeneity relative to non-African tumours.

2. Relating to line 209-210 - as clustering of Figure 2b was done specifically on the 861 ancestry-DMPs it is unsurprising that the samples in this heatmap do not cluster on geography. To identify geography DMPs a separate analysis using geography as the variable of interest would be required. It is also not clear how Figure 2b shows the stated 'extensive African-specific tumour heterogeneity' - The methylomes for each probe (row) in this plot look very uniform within the samples of men with African ancestry (as would be expected of a probe identified as an ancestry-DMPs).

Response: We agree that clustering performed on the 861 ancestry-associated DMPs would inherently reflect differences by ancestry rather than geography, given the design of the analysis. This heatmap was not intended to demonstrate a lack of geographic influence, but rather to visualise the extent to which the selected ancestry DMPs stratify individuals based on ancestral background. Regarding the statement on African-specific tumour heterogeneity, we appreciate that the methylation profiles at ancestry DMPs appear consistent within African samples in the heatmap and that the heterogeneity referenced is not captured in this probe-level heatmap. We have clarified this in the main text.

3. There is little overlap between African DMPs and TCGA DMPs - could this again be due to purity differences between the 2 cohorts. Do African DMPs trend in the same direction in the TCGA data? This could be assessed by a correlation of delta beta (tumour-normal methylation differences) between each dataset.

Response: We thank the reviewer for this observation and welcome the opportunity to clarify. Contrary to the impression of minimal overlap, the vast majority of DMPs identified in the African tumour-normal comparison do overlap with tumour-associated DMPs identified in TCGA dataset (Fig. 4h). Of the significant and validated tumour-associated DMPs and DMRs identified in African samples, only 405 did not overlap with those detected in TCGA and were therefore classified as "African-specific" for downstream analysis. This indicates that ~92% of African tumour-normal DMPs are shared with TCGA, highlighting a strong concordance in tumour-associated methylation profiles across datasets. We performed a correlation analysis of $\Delta\beta$ values (tumour-normal methylation differences) between our dataset and TCGA, which demonstrated strong directional concordance ($r = 0.86$, $p < 2.2 \times 10^{-16}$, 99.98% showed consistent direction of methylation change). The manuscript has been updated to include this result.

4. There is a surprising lack of overlap between the discovery and validation analyses in Figure 2g. What could explain this? Do the 861 DMPs trend in the same direction but not reach statistical cut-offs? Given the smaller sample size in the validation cohort, would a targeted analysis of just the 861 DMPs (reducing the multiple testing burden) provide a better assessment of the methylation change at these loci in the validation cohort.

Response: We thank the reviewer for this important point and have explored the replication of the 861 ancestry-associated DMPs in the validation cohort in more detail. Among the 861 discovery DMPs, 754 were present in the validation dataset (lack of total overlap due to independent probe filtering in each cohort). Of these, 207 (27.5%) reached nominal significance ($p < 0.05$), while 27 (3.6%) passed both the significance threshold and a $\Delta\beta \geq 20\%$

cutoff (consistent with our biological effect threshold). These 27 probes were used in downstream GSEA. While the overall Pearson correlation of $\Delta\beta$ values across all overlapping DMPs was low ($r = 0.12$), we note that 100% of the 27 validated DMPs and 96.4% of the 195 validated DMRs used in downstream analyses showed consistent direction of effect between discovery and validation cohorts.

We agree that reducing the multiple testing burden is a useful strategy to improve sensitivity in smaller or confounded validation cohorts, and it did help prioritise reproducible loci. However, the lack of tumour purity adjustment in the validation cohort, due to collinearity with ancestry, likely limited broader replication. In contrast, with the discovery cohort filtered for high tumour purity and adjusted accordingly, we believe this enabled more robust identification of ancestry-associated methylation differences. These findings support the conclusion that while full replication across all DMPs is limited, the subset of biologically meaningful and statistically robust loci used for downstream analyses exhibit strong reproducibility in both magnitude and direction of effect.

Specify experiments or analyses required to demonstrate the conclusions

- **Motivate your critique with relevant citations and argumentation**

1. Further work is needed to demonstrate that the ancestry methylation differences are not just differences in tumour purity. The best way to address this would be to include the methylation data from the samples that were excluded from the SAPCS cohort due to low tumour purity in all analyses.

Response: Please see earlier response on page 3-4.

2. Another approach would be to look specifically at the top probes in table 2 and perform correlation analysis and visualisations to compare their methylation levels with tumour purity across all tumour samples in the current study. The few European and Asian samples with high tumour purity could be highlighted in scatter plots or box plots to show that the difference is truly between ancestry groups and not just driven by tumour purity. Likewise, the boxplots in supplementary figure 6 could be repeated but with the data grouped by tumour purity.

Response: We thank the reviewer for this suggestion. In response, we performed correlation analyses and visualised methylation levels at the top 10 differentially methylated positions (DMPs) from Table 2, plotted against tumour purity across all prostate tumour samples. In the resulting scatterplots (Appendix Fig S7), samples are coloured by ancestry, and high tumour purity non-African samples are outlined to allow direct comparison across ancestries at matched purity. Several DMPs demonstrate clear ancestry-specific methylation differences that persist across the tumour purity spectrum, and high-purity non-Africans (though admittedly limited in number) cluster with their respective ancestry group rather than converging toward African methylation profiles, supporting the conclusion that these differences are not solely driven by tumour purity. We also generated corresponding boxplots (Appendix Fig S8) stratified by tumour purity quartile and ancestry. These show that methylation differences are generally maintained across multiple tumour purity strata, with several DMPs exhibiting consistent African/non-African divergence even outside of “TP Q4”. The top 10 DMRs showed more variable separation (Appendix Fig S9-S10).

Appendix Fig S5 (previously Supp Fig. 6) has also been updated – please check to see if I correctly interpreted your suggestion.

3. I appreciate that in some cases cell composition measures have been used in the regression models to identify DMPs, however is their use statistically valid given that they are highly correlated with the ancestry variable? Please discuss.

Response: We recognise that current statistical methods cannot perfectly resolve ancestry from correlated variables including cell content. Our analyses therefore represent the most rigorous approach currently feasible given the available data, though we note possible tumour purity confounding as a limitation.

4. To identify whether ancestry DMPs are tumour specific the normal prostate tissue methylation data should be added to the heatmap in figure 2b.

Response: Fig. 2b has been updated and confirms the tumour-specific nature of ancestry DMPs.

5. A separate analysis of the geographic difference within men of African ancestry would be needed to show that ancestry differences are more substantial than geographical differences. This would also help to address the question of whether environmental effects would be playing a role in prostate cancer, as suggested in the introduction.

Response: We thank the reviewer for this suggestion. However, in our dataset, ancestry and geography are completely confounded: all African men were recruited from South Africa, Asian men from Australia, and European men from South Africa. As such, a within-ancestry geographic comparison is not possible. We agree that this is an important question, and future studies including ancestry-matched individuals from different geographic regions will be needed to disentangle ancestry and environmental influences on prostate tumour methylation.

Minor points

- **Easily addressable points**

1. It would be helpful to point out that PC2 in the y-axis in Fig 1f only represents 5% of the variance in the dataset - so although there is a clear batch effect due to the different versions of the microarrays used, it is a relatively small source of variance compared to ancestry in PC1 - i.e. the main source of variance in the dataset is driven by biological rather than technical factors.

Response: Please see the y-axis of Fig. 1f, the PC % contribution has been noted. We thank the reviewer for this suggestion and have updated the manuscript text to reflect this point.

2. Why were different Beta value cut-off thresholds used for difference analysis - i.e. 20% for ancestry analysis; 30% for tumour-normal analysis; 10% for TCGA tumour-normal analysis. Was it for practical reasons such as having a manageable number of probes for downstream analysis in each case? Please also explain why different combinations of

covariates were used for different analyses (age, tumour purity, stromal and immune cell content).

Response: The thresholds for $\Delta\beta$ used across analyses were tailored for analytical practicality, as suggested. For the ancestry and tumour-normal analyses, different thresholds were applied to capture robust methylation shifts while enabling manageable downstream gene and pathway annotation. For TCGA-based comparison (450k), a 10% $\Delta\beta$ threshold was applied to maximise inclusion of probes given the smaller shared probe space with EPIC and the intent to broadly screen for overlap. For covariate selection, models were adjusted to include only variables that could be statistically justified and did not exhibit collinearity with the main variable of interest.

3. In cases where groups are unbalanced it would be preferable to use Wilcoxon tests rather than t-tests, e.g. for the analysis accompanying supp figure 6.

Response: Thank you for the suggestion, Supplementary Fig. 6, Figure 4b and Supplementary Fig. 14 (i.e. Appendix Fig S5, Fig 4B, Appendix Fig S17) have been updated accordingly.

Presentation and style

NA - figures are beautifully presented.

Response: Appreciate the acknowledgement.

Trivial mistakes

1. Figure 4d - is looks like the key label for cancer status is the wrong way around. i.e. the dark purple should be PCa.

Response: Thank you – the figure has been corrected.

Reviewer #2

Major Strengths

1. Really nice to see more research into non-European cohorts, definitely a strength of this paper, the lack of research into more ethnically diverse cohorts has been a long-standing weakness in cancer research to date.
2. Rigorous Technical Methodology. The authors demonstrate great attention to technical rigor in their analysis. Clear sense of the efforts of the research team to ensure all boxes were ticked and controlled for. Cross platform analysis, controlling for SNPs, comments on Type I vs Type II content - all great to see.

Response: Appreciate the acknowledgement.

Major Weaknesses

1. Predominantly Descriptive Analysis with Limited Biological Contextualization

While the technical analysis is comprehensive and methodologically sound, the manuscript suffers from what seems to be an overemphasis on observational descriptions, with deeper context or meaning frequently lacking, regarding biological interpretation or mechanistic insight, and/or the relevance pertaining to why some results are being commented on.

For example (pg 19, lines 340-344).

"tumour associated DMPs were primarily located in non-CGI regions (72.21%), followed by CGIs and shores (21.76%), while showing predominant enrichment in heterochromatin (34.12%, $p = 4.33 \times 10^{-13}$), followed by promoters (30.88%), including active non-prostate (12.16%, $p < 2.2 \times 10^{-16}$) and prostate lineage-specific (2.21%, $p < 2.2 \times 10^{-16}$), and bivalent poised (16.51%, $p < 2.2 \times 10^{-16}$) promoters, and enhancers (24.26%, $p = 2.51 \times 10^{-12}$)".

While the observations above are completely valid, my question is - what is the relevance of these findings? Why are the authors mentioning bivalent promoters specifically here (just as an example)? If it's being mentioned in this very granular list of genomic loci, it suggests this result should somehow relate to key differences underscoring ancestry differences in PCa. But bivalent promoters (as an example) aren't mentioned anywhere in the discussion, nor expanded upon elsewhere in the analysis.

Which begs the question - what is the key message, and why list so much superfluous information?

This is a frequent weakness in bioinformatically based papers, and my recommendation is always the same: if the information mentioned in results is just listing the top X number of statistically high hits from an intersection, but that information doesn't relate to the key conclusions of that experiments, it doesn't enhance understanding for the reader about the biological question under investigation.

This is just one example, so the authors shouldn't focus necessarily on rebutting this specific section I've highlighted. But throughout the manuscript results are presented in very minute detail, and in each case I would challenge to ask - **why** is this relevant to my understanding?. **How** do the results deeper our understanding of the underlying biology of PCa, ancestry, and DNA methylation.

Similarly, the discussion tends towards some general comments about broad pathways, but frequently fails to delve into the specifics within their own dataset.

For example, from DISCUSSION pg 26 lines 494-500:

"Analogous to our ancestry associations, African-derived tumours over non-tumours were primarily hypermethylated, with DMPs biased to non-CGI regions and displaying substantial enhancer distribution. Enhancers regulate gene expression distally via interactions with TFs and transcriptional machinery, and DNA looping to achieve proximity with gene promoters. Cancer significantly alters enhancer methylation, affecting (cancer-associated) target gene expression²⁵. We again observed target gene enrichment for "CHANDRAN_METASTASIS_DN", indicating both direct gene dysregulation through aberrant methylation and indirect effects via epigenetically altered interacting enhancers."

The section starts by making a broad comment about their data ("non-tumours were primarily hypermethylated") but the immediately moves into a commentary about enhancers that is so broad, it could relate to any dataset.

As the paper is framed in the context of ancestry, (and is the main innovation of the paper) a deeper integration of their results in the discussion is required, with references back to the main figure in the results so a reader can identify the data being referred to and it's relevance.

Response: We appreciate the reviewer's critique regarding the need for clearer biological contextualisation of our findings. While our original intention was to present a comprehensive and transparent account of our observations, we acknowledge that, at times, the manuscript may have over-emphasised descriptive detail at the expense of interpretive clarity. In response, we have carefully revised key sections of the Discussion to more clearly articulate the implications of key findings in relation to prostate cancer biology, ancestry and tumour progression.

2. Absence of Technical Validation Using Orthogonal Methods

There should be some technical validation of DMRs using an orthologous method, for example by targeted bisulfite resequencing. Given the differences observed between array probe types, the smaller cohort size, and other sources of variation and error (i.e., lot variations in the arrays), validation of their results by an orthologous method is required.

As a wet lab method, DNA methylation analysis using targeted resequencing is relatively easy and scales well to working with 2-3 plates of samples, and would be cost effective with a small panel of 50-100 regions (e.g., a single Nextseq 2000 Mid kit run using 1x150bp reads). While some validation with other publicly available data was done, technical validation on the same samples is needed to confirm the key findings regarding differences in DNA methylation. There's a tendency of the field to take beta values from the arrays as gospel, but the technical precision of the beadchip arrays isn't great, and I've seen firsthand how methylation values from arrays don't hold up when validated by sequencing - a problem of some significance when sample sizes get smaller (as they start to in this manuscript).

Response: We absolutely appreciate the reviewers' comments and as such we have already initiated a project to address this very question. However, we have forged a different approach where we are using long-read (ONT) and alternative targeted whole genome approaches (TWIST) to interrogate for a more complete spectrum of African-relevant DNA methylation sites on a subset of our African samples, while providing a source for EPIC array (all 800K CpG site) matched validation. Our reasoning for this approach is value of data gained for ever so precious and limited DNA derived from single fresh cores, which have already been used for alternative whole genome interrogations, as well as for multiple samples EPIC v1 and v2 repeated data generation. As such, our sample source is depleted for a targeted validation approach. Besides **technical validations**, the team is currently recruiting additional patients to ultimately provide **cohort-matched validation data**. This cohort will still take some time to recruit and as such does not form part of this study.

Minor Comments

1. Would suggest also adding in a comment that the variation could be attributable to lot

differences in array version (pg5 lines 129-135). Historically for our EPIC studies we try to request Illumina ship arrays from the sample lot to control for this.

Response: Thank you, we have updated the manuscript text accordingly.

2. Conservative Approach to Sample Reclassification Recommended

I'm not enthusiastic about the decision of the authors to reclassify 7 non-tumour to "presumably PCa" based on methylation data alone, since it's something of a logical leap to discount the pathology histopathology classification based entirely base just on the methylation data.

It would have been better to exclude them fully from the analysis, or at least analyze them separately. My reading of the paper is that they were lumped in with the rest of the PCa samples, and it's difficult to infer what effect these 7 samples could have, particularly when numbers start to dwindle on the subgroup analyses.

I won't request re-analysis, but the authors needs to revise Table1 and show how many samples in a subgroup contain these 7 outliers (and also note this elsewhere in other figures, when this 7 are part of the analysis set).

For example, the Ancestry-associated cohort has:

Discovery = 21

Validation = 35

The table needs to be revised to show (for example, just making up numbers):

Discovery = 21 (6 = presumably PCa)

Validation = 35 (1 = presumably PCa)

Response: We thank the reviewer for this comment and acknowledge the concern regarding reclassification of the seven samples as "presumably PCa". We have revised Table 1 and the relevant figures to clarify their inclusion.

3. Their figure6 is a nice snapshot of the key conclusions, but the figure title needs to be changed to make it clear this is the author's interpretation and proposed model, as there's no functional data in their study that definitively establishes this is mechanistically what is happening. Simply revising to say "Proposed Model....etcetc" would be sufficient.

Response: We have updated the title for Figure 6.

Recommendation

Work that is both interesting to the field and relevant, but the current manuscript has some notable gaps as outlined above.

Response: We appreciate the reviewers' comments and have addressed all concerns, although we are unable at this time to generate the required technical validation due to sample exhaustion.

Given my recommendation to generate supporting validation data, either major revision, or opportunity to resubmit, would both be appropriate.

Response: We appreciate the opportunity to resubmit.

Reviewer #3

The authors are addressing a key gap in cancer omics research: the lack of data covering the great heterogeneity of African populations. With a rather large sample size (nearly 200), this dataset will be a great resource for future prostate cancer (PCa) studies, and will probably be cited heavily.

Response: Appreciate the acknowledgement that this is a large sample size and the first of its kind for the research community, creating we envisage broad interest in the resource.

The analysis is as rigorous as the current state of the art allows. The authors spent significant efforts identifying confounding CpGs in the methylation chip, and systemically assessed the confounding effects of 2 versions of the chip. They also included tumor purity information in the majority of figures, making confounding factors very obvious.

That being said, the conclusions are still confounded by tumor purity. However, this is not a limitation of this manuscript per se, but a limitation of the field as it is. When comparing tumor to non-tumor, for example, non-tumor samples will inevitably have higher stromal fractions. I appreciate that the authors were very transparent about this. Although a few deconvolution methods have been developed that aim to computationally split the data into tumor-only and stromal-only, none of them seem to work very well from my experience. In short, this limitation should not diminish the quality and value of this dataset itself. It just means that this dataset will be reanalyzed in the future once better methodologies have been developed.

A related discussion is whether it is appropriate to filter African PCa samples to only use those with high tumor purity. This selection procedure induces bias since African samples will have higher purity than European/Asian samples. However, without selection, the comparison would be "impure samples vs impure samples", which is not meaningful. The authors acknowledged the bias induced by the selection, which is rigorous. Importantly, they were able to observe that the great heterogeneity of African samples can not be explained by purity.

Response: Again, we appreciate the recognition of our efforts placed on array version verification (technical repeats) and confounding analyses. As with Reviewer 1, this reviewer also raises concerns with regards to confounding due to tumour purity, which we have addressed extensively under responses to Reviewer 1 – please kindly consult those responses.

Major revisions

1. Visually, Fig. 2b doesn't correlate well with Fig. 2c. Fig. 2c shows that the absolute methylation differences of the DMPs are at least 0.2, but this is not visible for the majority of rows shown in Fig. 2b. This could be due to the color scale. It is important to adjust this since the heatmap helps the reader visualize the consistency/heterogeneity within each

ancestry. One possible method is to include a row-scaled z-score-based heatmap side-by-side.

Response: We agree and thank the reviewer for the suggestion – Fig. 2b has been updated to include the z-score-based heatmap.

2. In Fig. 2g, the validation cohort showed many more DMPs and DMRs (16 times more DMPs, for example). The authors need to explain how this key inconsistency occurred. The authors also need to calculate if such overlap is significant over a random background, given the large number of DMP/DMRs in the validation cohort. In addition, the Venn plot is misleading since the area of overlap does not reflect the percentage of overlap.

Response: The greater number of ancestry-associated DMPs and DMRs identified in the validation cohort likely reflects two key factors. The first being that we were unable to adjust for tumour purity in the validation dataset due to its complete collinearity with ancestry, so we would expect increased detection of tumour purity-associated differences rather than ancestry-specific signal. Second, the validation cohort is derived from the EPICv2 array, which has broader CpG coverage than EPICv1 and therefore allows detection of additional loci not captured in the discovery cohort.

To evaluate whether the overlap between discovery and validation datasets exceeds random expectation, we performed permutation-based enrichment tests using matched probe sets, as suggested. For DMPs, the observed overlap of 27 probes was significantly greater than expected by chance ($p = 0.0023$). For DMRs, the observed overlap of 195 DMR CpGs yielded a permutation p -value <0.0001 , confirming that the overlap is highly unlikely to have occurred by chance and supports the reproducibility of ancestry-associated differential methylation across platforms. We've revised the manuscript to include this result.

Regarding the visual representation in Fig. 2g, we have amended the figure to include area-proportional Venn diagrams to more accurately represent the size of the overlapping and unique probe sets. We have also updated Fig. 4h.

3. For the key genes reported in the manuscript (*GALM*, *EVC2*, *CHSY1*, and *SPDYA*; *PYCARD* and *SLC12A9*), additional validation is needed to ascertain how much they are confounded by tumor purity. The authors could take advantage of public datasets (if any) that profiled the methylation levels of these loci in purified luminal, stromal, and immune cells.

Response: We thank the reviewer for this suggestion. Given the strong ancestry component of our findings and the known limitations of public reference datasets (which are largely European-biased), we believe that integrating such external data could introduce interpretive bias, especially for loci potentially specific to African populations. Instead, we directly evaluated the relationship between methylation and tumour purity at the four key ancestry-associated loci (*GALM*, *EVC2*, *CHSY1*, *SPDYA*) by plotting β -values against tumour purity, and by visualising ancestry-stratified methylation distributions across tumour purity quartiles (Figure EV3). These plots demonstrate that ancestry-associated methylation differences become more pronounced with increasing tumour purity, and are consistently observed across purity strata, suggesting that the observed divergence is unlikely to be solely purity-driven. For *PYCARD* and *SLC12A9*, which were identified from tumour versus normal comparisons, confounding by tumour purity is less relevant given the nature of the comparison.

Minor issues

1. Fig. 1a-d needs an explicit title. It feels like the distribution of all probes, not the probes that are confounded.

Response: Thank you for the suggestion. We have provided an explicit title.

2. Line 210: "extensive African-specific tumour heterogeneity (Fig. 2b)". Not sure what this is referring to. I did not observe any heterogeneity in Fig. 2b

Response: Please see our responses to Reviewer 1, where we have addressed this.

Pre-decision cross-commenting

Reviewer #3

As the more "optimistic" reviewer #3, I do understand the points made by reviewer #1. Two approaches to address this issue were brought up: 1) reanalyze the data with all samples (including the impure ones) and 2) for the top discoveries listed in Table 2, rule out the possibility that they are confounded by purity.

I strongly favor the second approach. Although approach 1) will lead to a more apples-to-apples comparison, interpreting the results might be challenging, since it will be impure African samples vs impure European/Asian samples. By contrast, approach 2) is critical, since Table 2 contains the key findings of this paper that could be cited in the field widely, necessitating extra efforts to make sure they are not strongly confounded. I also raised a similar point in Major Point 3 for the key genes reported in the manuscript (GALM, EVC2, CHSY1, and SPDYA; PYCARD and SLC12A9).

Reviewer #1

I agree with reviewer 3's new comment that additional analysis of the top discoveries listed in Table 2 is a good compromise, and thus I still recommend the straightforward analyses recommended in my original review. i.e. 'look specifically at the top probes in table 2 and perform correlation analysis and visualisations to compare their methylation levels with tumour purity across all tumour samples in the current study. The few European and Asian samples with high tumour purity could be highlighted in scatter plots or box plots to show that the difference is truly between ancestry groups and not just driven by tumour purity. Likewise, the boxplots in supplementary figure 6 could be repeated but with the data grouped by tumour purity.

Response: We appreciate the reviewers' suggestions and believe our responses in this document and our revised manuscript address their concerns and requests.

Editors' overall comments for consideration

Both Reviewers #1 and #3 raised concerns about potential confounding due to tumor purity. During our pre-decision cross-commenting process (in which editors invite reviewers to make additional comments on specific points), they offered constructive suggestions for addressing them. These additional comments are included below following their individual reports.

Reviewer #2 had concerns regarding the presentation of the manuscript and requested a better integration of the results in the discussion. They also requested additional technical validation using orthogonal methods. These points should be carefully addressed.

Response: These points have been carefully addressed.

2nd Sep 2025

Manuscript Number: MSB-2025-13061R

Title: Methylation reprogramming associated with aggressive prostate cancer and ancestral disparities

Author: Jenna Craddock

Pavlo Lutsik

Pamela Soh

Melanie Louw

Md. Hasan

Sean Patrick

Shingai Mutambirwa

Phillip Stricker

Hagen Förtsch

MS Bornman

Clarissa Gerhauser

Vanessa Hayes

Dear Prof Hayes,

Thank you for submitting the revised version of your manuscript. We have now received feedback from the two reviewers who agreed to evaluate the revision.

As you will see below, both reviewers are overall satisfied with the changes made. However, Reviewer #1 has raised several minor concerns that we ask you to address in a further revision of the manuscript.

1. Regarding Reviewer #1's comment #5, we have received additional input from both reviewers during the cross-commenting phase, which is included below after the review reports. In light of these comments, please provide a new response accordingly.
2. If feasible and in accordance with individual consent agreements, please deposit the full raw datasets in a public repository prior to your next submission. Be sure to update the Data Availability section to reflect this.
3. Other concerns need to be addressed in writing.

On a more editorial level, please do the following:

1. Remove the "Author contribution statement" section from the manuscript file.
2. Ensure that all figure callouts are listed sequentially. Please provide missing callouts for Fig. 6A-C
3. The section titled "Competing interests" should be renamed to "DISCLOSURE AND COMPETING INTERESTS STATEMENT".
4. The code availability should be merged with the Data availability, and the heading "code availability" should be removed. Please remove the reviewer token and make sure the datasets are made publicly available upon the acceptance of the manuscript.
5. Please upload all EV figures as individual, high-resolution figure files. The legends for EV figures should stay in the main manuscript file.
6. Appendix
 - Table of Content should list each individual Appendix figure along with its corresponding page number.
 - Remove the "list of Appendix figures", "List of Expanded View Tables", and "Description of Expanded View datasets" from the Appendix pdf.
7. EV tables and datasets
 - Since the tables are quite complex, please convert all of them to Expanded View (EV) datasets and update the corresponding callouts in the manuscript file accordingly.
 - Legends should be provided as a separate tab/sheet within each Excel file, rather than being included in the Appendix or submitted as a zipped README file.
 - Each dataset should be uploaded as an individual .xlsx file, not as a compressed (zipped) folder.
8. Please address the following issues related to figure legends:

- Please note that the exact p values are not provided in the legend of figure EV2
- Please indicate the statistical test used for data analysis in the legends of figures 2C, G; 4C, I
- Please note that the box plots need to be defined in terms of minima, maxima, centre, bounds of box and whiskers, and percentile in the legends of figures 4B
- Please note that information related to n is missing in the legends of figures 2C, G; 3, 4C

Click on the link below to submit your revised paper.

Sincerely,
Jingyi

Jingyi Hou, PhD
Senior Editor
Molecular Systems Biology

*** PLEASE NOTE *** As part of the EMBO Press transparent editorial process initiative (see our Editorial at <https://dx.doi.org/10.1038/msb.2010.72> , Molecular Systems Biology will publish online a Review Process File to accompany accepted manuscripts. When preparing your letter of response, please be aware that in the event of acceptance, your cover letter/point-by-point document will be included as part of this File, which will be available to the scientific community. More information about this initiative is available in our Instructions to Authors. If you have any questions about this initiative, please contact the editorial office (msb@embo.org).

Reviewer #1:

Major points

1. I disagree that the change in PC associations after filtering is evidence that the filtering has 'reduced non-biological variability and enhanced detection of ancestry-associated methylation differences' - this change could be due to the fact that confounding between ancestry and cellular composition has been increased by the filtering step (so cellular differences now appear as ancestry changes). That being said, I understand that there is no perfect solution with the current dataset. I appreciate the steps taken to include tumour purity as a covariate in models, and the new plots showing the methylation levels of non-African high tumour purity samples are convincing that (at least) the top loci are not driven by purity.

The argument that immune contribution to variance is relatively low, and therefore immune cell composition is unlikely to be driving the ancestry associated methylated signal is reasonable.

I would still question whether there is enough evidence to claim that African tumours are more heterogeneous. The authors use the fact that tumours filtered to Q4 are still dispersed as evidence that heterogeneity is independent of cell composition, yet Q4 tumours still comprise a broad range of T-luminal cell percentages from 40-100% (whereas most non-African tumours are from a narrower range of 0-40% T-luminal cells). This could likewise explain the difference in variance found by Levene's tests. I think language should be softened around the heterogeneity claim in the discussion.

2. Thank you for the clarification and change in text

3. Thank you for the clarification about the overlap between African DMPs and TCGA DMPs

4. The explanation as to the lack of reproducibility between datasets makes sense, and supports use of the 27 DMPs as being the most robust changes.

Specific experiments or analyses

1. The plots are a useful addition to the paper and show that the top 10 ancestry DMPs are not solely driven by purity. This is very reassuring. Yes my suggestion for figure S5 has been correctly interpreted.

4. The addition of normal prostate tissue methylation data is helpful and shows that normal African tissue is distinct from European tumour tissue, which helps allay the concern that ancestry DMPs are driven by purity.

5. The authors respond that 'in our dataset, ancestry and geography are completely confounded' and a within-ancestry geographic comparison is not possible' yet in the paper state that 'Our findings highlight ancestry over geographic-associated prostate tumour methylation, greater African-specific heterogeneity and point towards ancestry-informed biomarkers.' How can they make this claim if geography and ancestry are confounded?

All minor comments have been adequately addressed.

Recommendation:

The GEO dataset only contains the samples that survived QC. Typically, authors would include the full raw dataset, such that other researchers would be able to follow the methods and reproduce all analysis steps from scratch. With the ethos of reproducible research, and given the concern of 2 reviewers about the confound of sample purity with ancestry, I would recommend upload of the n=93 excluded low purity samples on GEO - this way other researchers can apply their own methods to attempt to deconvolute the data and make further use of the valuable resource that the authors have created. Alternatively the authors could make this data 'available on request'. However I will leave this to the editors' discretion.

Reviewer #3:

All of my major concerns have been resolved. I have no further questions.

Referee cross-commenting

Reviewer #3

To me, the statement "Our findings highlight ancestry over geographic-associated prostate tumour methylation" at Line 387 is fine. The prostate cancer samples from African and European ancestries are both from South Africa, and yet they are highly different. This is supported by Fig. 2a, where the samples segregate on PCA plot mostly by ancestry (African vs non-African), not by geography (South Africa vs Australia). I do think the rebuttal that "ancestry and geography are completely confounded" is confusing, but the main text seems fine.

Reviewer #1

Yes, it would be true to say that just comparing European ancestry South Africans versus African ancestry South Africans the methylation differs by ancestry

Repose to Editor – R3

Please note that the Datasets have not been uploaded correctly and we need to clarify all the Excel files uploaded to our submission system:

1. There are 6 Datasets uploaded as individual files and a zip folder "EV tables EV1 to EV30" - could you confirm that all these files are from the previous submission? If that is the case, please remove them. If not, please contact us for more instructions how to upload them correctly.

Response: They were from the previous submission – apologies, and have been removed.

2. There is a zip folder "Datasets (zipped)" with 36 datasets that are completely correct (source name, labels and legends, and callouts), but should be uploaded as individual Data Set files, instead of being zipped

Response: Zip file removed and individual dataset files uploaded. Dataset EV1 to EV36.

3. Thank you for providing the Synopsis image, however, it is too large and the ratio is not correct. Please make sure to re-arrange the synopsis image into the appropriate proportions I tried resizing the image and the ratio is 550x778. The synopsis image should be in .jpg or .png format, exactly 550 pixels wide and 300-600 pixels high (the height is variable). Please note that the text needs to be legible at the final size.

Response: The image appears to be the correct size. I have now uploaded the .png version to see if this works. I have also uploaded the license to publish as this figure and Figure 6 were created using BioRender. Both licenses uploaded.

17th Sep 2025

Manuscript number: MSB-2025-13061RR

Title: Methylation reprogramming associated with aggressive prostate cancer and ancestral disparities

Dear Prof Hayes,

Thank you again for sending us your revised manuscript. We are now satisfied with the modifications made and I am pleased to inform you that your paper has been accepted for publication.

Sincerely,
Jingyi

Jingyi Hou, PhD
Senior Editor
Molecular Systems Biology
